# Improving Generalization in Meta-RL with Imaginary Tasks from Latent Dynamics Mixture

**Suyoung Lee**[*]
KAIST
suyoung.l@kaist.ac.kr

**Sae-Young Chung**
KAIST
schung@kaist.ac.kr

## Abstract

The generalization ability of most meta-reinforcement learning (meta-RL) methods is largely limited to test tasks that are sampled from the same distribution used to sample training tasks. To overcome the limitation, we propose Latent Dynamics Mixture (LDM) that trains a reinforcement learning agent with imaginary tasks generated from mixtures of learned latent dynamics. By training a policy on mixture tasks along with original training tasks, LDM allows the agent to prepare for unseen test tasks during training and prevents the agent from overfitting the training tasks. LDM significantly outperforms standard meta-RL methods in test returns on the gridworld navigation and MuJoCo tasks where we strictly separate the training task distribution and the test task distribution.

## 1 Introduction

Overfitting and lack of generalization ability have been raised as the most critical problems of deep reinforcement learning (RL) [5, 8, 30, 34, 38, 47, 52]. Numbers of meta-reinforcement learning (meta-RL) methods have proposed solutions to the problems by meta-training a policy that easily adapts to unseen but similar tasks. Meta-RL trains an agent in multiple sample tasks to construct an inductive bias over the shared structure across tasks. Most meta-RL works evaluate their agents on test tasks that are sampled from the same distribution used to sample training tasks. Therefore, the vulnerability of meta-RL to test-time distribution shift is hardly revealed [12, 26, 29, 30].

One major category of meta-RL is gradient-based meta-RL that learns an initialization of a model such that few steps of policy gradient are sufficient to attain good performance in a new task [9, 36, 39, 40, 55]. Most of these methods require many test-time rollouts for adaptation that may be costly in real environments. Moreover, the networks are composed of feedforward networks that make online adaptation within a rollout difficult.

Another major category of meta-RL is context-based meta-RL that tries to learn the tasks' structures by utilizing recurrent or memory-augmented models [6, 13, 23, 25, 32, 35, 49, 56]. A context-based meta-RL agent encodes its collected experience into a context. The policy conditioned on the context is trained to maximize the return. These methods have difficulties generalizing to unseen out-of-distribution (OOD) tasks mainly because of two reasons. **(1)** The process of encoding unseen task dynamics into a context is not well generalized. **(2)** Even if the unseen dynamics is well encoded, the policy that has never been trained conditioned on the unseen context cannot interpret the context to output optimal actions.

We propose Latent Dynamics Mixture (LDM), a novel meta-RL method that overcomes the aforementioned limitations and generalizes to strictly unseen test tasks without any additional test-time updates. LDM is based on variational Bayes-adaptive meta-RL that meta-learns approximate inference on a latent belief distribution over multiple reward and transition dynamics [56]. We generate imaginary

---

[*]Corresponding author.

35th Conference on Neural Information Processing Systems (NeurIPS 2021).

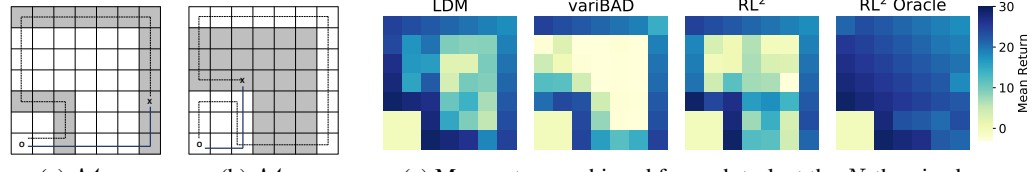

(a) $\mathcal{M}_{\text{train}}$        (b) $\mathcal{M}_{\text{test}}$        (c) Mean returns achieved for each task at the $N$-th episode.

Figure 1: The gridworld example for the problem setup. **(a)** Training MDPs $\mathcal{M}_{\text{train}}$: a goal is located at one of the 18 shaded states. During training, the agent has to navigate to discover the unknown goal position randomly sampled at the beginning of each task. **(b)** Test MDPs $\mathcal{M}_{\text{test}}$: a test goal is located at one of the 27 shaded states disjoint to the set of training goals in **(a)**. The agent does not have access to the tasks in $\mathcal{M}_{\text{test}}$ during training. The solid lines and dashed lines represent examples of optimal paths with and without the knowledge of the true goals, respectively. $\circ$ and $\times$ symbols represent the initial state and the hidden goal state, respectively.

tasks using mixtures of training tasks' meta-learned latent beliefs. By providing the agent with imaginary tasks during training, the agent can train its context encoder and policy given the context for unseen tasks that may appear during testing. Since LDM prepares for the test during training, it does not require additional gradient adaptation during testing.

For example, let there be four types of training tasks, each of which must move east, north, west, and south. By mixing the two tasks of moving east and north, we may create a new task of moving northeast. By mixing the training tasks in different weights, we may create tasks with goals in any direction.

We evaluate LDM and other meta-RL methods on the gridworld navigation task and MuJoCo meta-RL tasks, where we completely separate the distributions of training tasks and test tasks. We show that LDM, without any prior knowledge on the distribution of test tasks during training, achieves superior test returns compared to other meta-RL methods.

## 2 Problem Setup

Our work is motivated by the meta-learning setting of variBAD [56], therefore we follow most of the problem setup and notations except for a key difference that the test and training task distributions are strictly disjoint in our setup. A Markov decision process (MDP) $M = (\mathcal{S}, \mathcal{A}, R, T, T_0, \gamma, H)$ consists of a set of states $\mathcal{S}$, a set of actions $\mathcal{A}$, a reward function $R(r_{t+1}|s_t, a_t, s_{t+1})$, a transition function $T(s_{t+1}|s_t, a_t)$, an initial state distribution $T_0(s_0)$, a discount factor $\gamma$ and a time horizon $H$.

During meta-training, a task (or a batch of tasks) is sampled following $p(M)$ over the set of MDPs $\mathcal{M}$ at every iteration. Each MDP $M_k = (\mathcal{S}, \mathcal{A}, R_k, T_k, T_0, \gamma, H)$ has individual reward function $R_k$ (e.g., goal location) and transition function $T_k$ (e.g., amount of friction), while sharing some general structures. We assume that the agent does not have access to the task index $k$, which determines the MDP. At meta-test, standard meta-RL methods evaluate agents on tasks sampled from the same distribution $p$ that is used to sample the training tasks. To evaluate the generalization ability of agents in environments unseen during training, we split $\mathcal{M}$ into two strictly disjoint training and test sets of MDPs, i.e., $\mathcal{M} = \mathcal{M}_{\text{train}} \cup \mathcal{M}_{\text{test}}$ and $\mathcal{M}_{\text{train}} \cap \mathcal{M}_{\text{test}} = \emptyset$. The agent does not have any prior information about $\mathcal{M}_{\text{test}}$ and cannot interact in $\mathcal{M}_{\text{test}}$ during training.

Since the MDP is initially unknown, the best the agent can do is to update its belief $b_t(R, T)$ about the environment according to its experience $\tau_{:t} = \{s_0, a_0, r_1, s_1, a_1, r_2, \ldots, s_t\}$. According to the Bayesian RL formulation, the agent's belief about the reward and transition dynamics at timestep $t$ can be formalized as a posterior over the MDP given the agent's trajectory, $b_t(R, T) = p(R, T|\tau_{:t})$. By augmenting the belief to the state, a Bayes-Adaptive MDP (BAMDP) can be constructed [7]. The agent's goal in a BAMDP is to maximize the expected return while exploring the environment by minimizing the uncertainty about the initially unknown MDP.

The inference and posterior update problem in a BAMDP can be solved by combining meta-learning and approximate variational inference [56]. An inference model encodes the experience into a low-

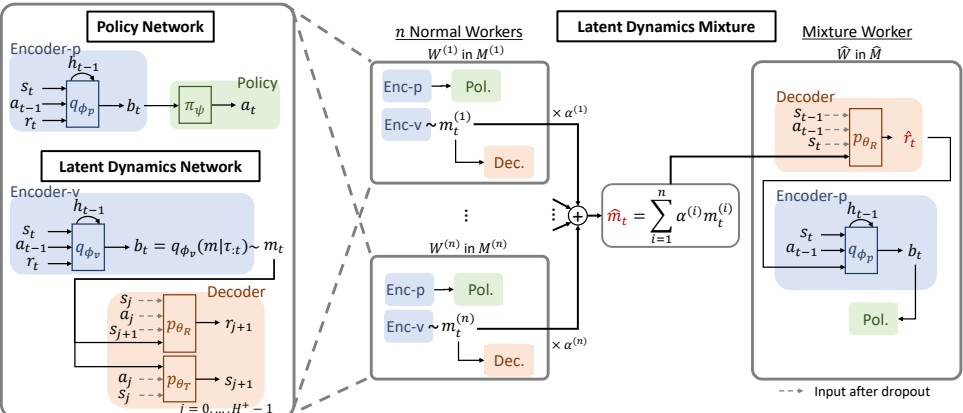

Figure 2: Imaginary task generation from latent dynamics mixture. We train $n$ normal workers and a mixture worker in parallel. Each normal worker $W^{(i)}$ trains a policy network and a latent dynamics network on its sampled MDP $M^{(i)} \in \mathcal{M}_{\text{train}}$. A mixture latent model $\hat{m}_t$ is generated as a weighted sum of normal workers' latent models $m_t^{(i)}$. All workers share a single policy network and a single latent dynamics network. We feed this mixture belief to the latent dynamics network's learned decoder to generate a new reward $\hat{r}_t$ and construct an imaginary task $\hat{M}$.

dimensional stochastic latent variable $m$ to represent the posterior belief over the MDPs.[2] The reward and transition dynamics can be formulated as shared functions across MDPs: $R(r_{t+1}|s_t, a_t, s_{t+1}; m)$ and $T(s_{t+1}|s_t, a_t; m)$. Then the problem of computing the posterior $p(R, T|\tau_{:t})$ becomes inferring the posterior $p(m|\tau_{:t})$ over $m$. By conditioning the policy on the posterior $p(m|\tau_{:t})$, an approximately Bayes-optimal policy can be obtained.

Refer to Figure 1 for the gridworld navigation example that is the same as the example used in variBAD [56] except for the increased number of cells and that the task set $\mathcal{M}$ is divided into disjoint $\mathcal{M}_{\text{train}}$ and $\mathcal{M}_{\text{test}}$. A Bayes-optimal agent for a task in $\mathcal{M}_{\text{train}}$ first assigns a uniform prior to the goal states of $\mathcal{M}_{\text{train}}$ (Figure 1a) and then explores these states until it discovers a goal state as the dashed path in Figure 1a. If this agent, trained to solve the tasks in $\mathcal{M}_{\text{train}}$ only, is put in to solve a task in $\mathcal{M}_{\text{test}}$ without any prior knowledge (Figure 1b), the best policy an agent can take first is to maintain its initial belief learned in $\mathcal{M}_{\text{train}}$ and explore the goal states of $\mathcal{M}_{\text{train}}$. Once the agent realizes that there are no goal states in $\mathcal{M}_{\text{train}}$, it could start exploring the states that are not visited (i.e., $\mathcal{M} - \mathcal{M}_{\text{train}}$), and discover an unseen goal state in $\mathcal{M}_{\text{test}}$. However, it is unlikely that the agent trained only in $\mathcal{M}_{\text{train}}$ will encode its experience into beliefs for unseen tasks accurately and explore the goal states of $\mathcal{M}_{\text{test}}$ efficiently conditioned on the unseen context without any prior knowledge or test-time adaptation.

## 3 Latent Dynamics Mixture

Our work aims to train an agent that prepares for unseen test tasks during training as in Figure 1b. We provide the agent during training with imaginary tasks created from mixtures of training tasks' latent beliefs. By training the agent to solve the imaginary tasks, the agent learns to encode unseen dynamics and to produce optimal policy given the beliefs for tasks not only in $\mathcal{M}_{\text{train}}$ but also for more general tasks that may appear during testing.

Refer to Figure 2 for an overview of the entire process. We train $n$ normal workers $W^{(1)}, \ldots, W^{(n)}$ and a mixture worker $\hat{W}$ in parallel. For the convenience of explanation, we first focus on the case with only one mixture worker. At the beginning of every iteration, we sample $n$ MDPs $M^{(1)}, \ldots, M^{(n)}$ from $\mathcal{M}_{\text{train}}$ and assign each MDP $M^{(i)}$ to each normal worker $W^{(i)}$. All normal and mixture workers share a single policy network and a single latent dynamics network. Normal workers train the shared policy network and latent dynamics network using true rewards from the sampled MDPs. A mixture

---

[2]We use the terms context, latent belief, and latent (dynamics) model interchangeably to denote $m$.

worker trains the policy network with imaginary rewards from the learned decoder's output given mixture beliefs.

## 3.1 Policy Network

Any type of recurrent network that can encode the past trajectory into a belief state $b_t$ is sufficient for the policy network. We use an $RL^2$ [6] type of policy network (Figure 2). Each normal worker $W^{(i)}$ trains a recurrent Encoder-p (parameterized by $\phi_p$) and a feedforward policy network (parameterized by $\psi$) to maximize the return for its assigned MDP $M^{(i)}$. A mixture worker trains the same policy network to maximize the return in an imaginary task $\hat{M}$ where the imaginary reward $\hat{r}_t$ is from the decoder given the mixture model $\hat{m}_t$. Any online RL algorithm can be used to train the policy network. We use A2C for the gridworld and PPO [37] for MuJoCo tasks to optimize $\phi_p$ and $\psi$ end-to-end.

## 3.2 Latent Dynamics Network

We use the same network structure and training methods of VAE introduced in variBAD [56] for our latent dynamics network (Figure 2). The only difference is that the policy network and the latent dynamics network do not share an encoder. Therefore Encoder-v (parameterized by $\phi_v$) of the latent dynamics network does not need to output the context necessary for the policy but only needs to encode the MDP dynamics into a low-dimensional stochastic latent embedding $m$. The latent dynamics model $m$ changes over time as the agent explores an MDP (denoted as $m_t$), but converges as the agent collects sufficient information to infer the dynamics of the current MDP. The latent dynamics network is not involved in the action selection of workers. We store trajectories from $\mathcal{M}_{\text{train}}$ in a buffer and use samples from the buffer to train the latent dynamics network offline. Each normal worker trains the latent dynamics network to decode the entire trajectory, including the future, to allow inference about unseen future transitions. In this work, we focus on MDPs where only reward dynamics varies [9, 10, 12, 13, 17, 26] and only train the reward decoder (parameterized by $\theta_R$) as in [56]. The parameters $\phi_v$ and $\theta_R$ are optimized end-to-end to maximize the ELBO [20] using a reparameterization trick.

## 3.3 Imaginary Task Generation from Latent Dynamics Mixture

While training the policy network of the normal workers $W^{(1)}, \ldots, W^{(n)}$ in parallel, we generate an imaginary latent model $\hat{m}_t$ as a randomly weighted sum of the latent models $m_t^{(1)}, \ldots, m_t^{(n)}$ of the normal workers.

$$\hat{m}_t = \sum_{i=1}^{n} \alpha^{(i)} m_t^{(i)} \quad \text{and} \quad \alpha^{(1)}, \ldots, \alpha^{(n)} \sim \beta \cdot \text{Dirichlet}(1,\ldots,1) - \frac{\beta - 1}{n}. \tag{1}$$

$\alpha^{(i)}$'s are random mixture weights multiplied to each latent model $m_t^{(i)}$. At the beginning of every iteration when the normal workers are assigned to new MDPs, we also sample new mixture weights fixed for that iteration. There are many distributions suitable for sampling mixture weights, and we use the Dirichlet distribution in Equation 1. $\beta$ is a hyperparameter that controls the mixture's degree of extrapolation. The sum of mixture weights equals 1 regardless of the hyperparameter $\beta$. If $\beta = 1$, all $\alpha^{(i)}$'s are bounded between 0 and 1. Then the mixture model becomes a convex combination of the training models. If $\beta > 1$, the resulting mixture model may express extrapolated dynamics of training tasks. We find $\beta = 1.0$ suits best for most of our experiments among $\{0.5, 1.0, 1.5, 2.0, 2.5\}$ that we tried. Refer to extrapolation results in Section 5.1.2 where $\beta$ greater than 1 can be effective.

A mixture worker interacts with an MDP sampled from $\mathcal{M}_{\text{train}}$, but we replace the environment reward $r_t$ with the imaginary reward $\hat{r}_t$ to construct a mixture task $\hat{M}$. A mixture worker trains the policy network to maximize the return for the imaginary task $\hat{M}$. We expect the imaginary task $\hat{M}$ to share some common structures with the training tasks because the mixture task is generated using the decoder that is trained to fit the training tasks' reward dynamics. On the other hand, the decoder can generate unseen rewards because we feed the decoder unseen mixture beliefs. The mixture worker only trains the policy network but not the decoder with imaginary dynamics of $\hat{M}$.

**Dropout of state and action input for the decoder**    As the latent dynamics network is trained for the tasks in $\mathcal{M}_{\text{train}}$, we find that the decoder easily overfits the state and action observations, ignoring the latent model $m$. Returning to the gridworld example, if we train the decoder with the tasks in $\mathcal{M}_{\text{train}}$ (Figure 1a), and feed the decoder one of the goal states in $\mathcal{M}_{\text{test}}$, the decoder refers to the next state input $s_{i+1}$ only and always returns zero rewards regardless of the latent model $m$ (Figure 4a). We apply dropout of rate $p_{\text{drop}}$ to all inputs of the decoder except the latent model $m$. It forces the decoder to refer to the latent model when predicting the reward and generate general mixture tasks. Refer to Appendix E for ablations on dropout.

Training the decoder with a single-step regression loss is generally less complex than training the policy network with a multi-step policy gradient loss. Therefore the decoder can be stably trained even with input dropout, generalizing better than the meta-trained policy. Refer to Appendix B for empirical results on the test-time generalization ability of the latent dynamics network with dropout.

### 3.4    Implementation Details

**Multiple episodes of the same task in one iteration**    Following the setting of variBAD [56], we define an iteration as a sequence of $N$ episodes of the same task and train the agent to act Bayes-optimally within the $N$ rollout episodes (i.e., $H^+ = N \times H$ steps). After every $N$ episodes, new tasks are sampled from $\mathcal{M}_{\text{train}}$ for the normal workers, and new mixture weights are sampled for the mixture worker. Then we can compare our method to other meta-RL methods that are designed to maximize the return after rollouts of many episodes.

**Multiple mixture workers**    We may train more than one mixture worker at the same time by sampling different sets of mixture weights for different mixture workers. Increasing the ratio of mixture workers to normal workers may help the agent generalize to unseen tasks faster, but the normal workers may require more iterations to learn optimal policies in $\mathcal{M}_{\text{train}}$. We train $n = 14$ normal workers and $\hat{n} = 2$ mixture workers in parallel unless otherwise stated. Refer to Appendix D for empirical analysis on the ratio of workers.

## 4    Related Work

**Meta-Reinforcement Learning**    Although context-based meta-RL methods require a large amount of data for meta-training, they can learn within the task and make online adaptations. RL$^2$ [6] is the most simple, yet effective context-based model-free meta-RL method that utilizes a recurrent network to encode the experience into a policy. PEARL [35] integrates an off-policy meta-RL method with online probabilistic filtering of latent task variables to achieve high meta-training efficiency. MAML [9] learns an initialization such that a few steps of policy gradient is enough for the agent to adapt to a new test task. E-MAML and ProMP [36, 40] extend MAML by proposing exploration strategies for collecting rollouts for adaptation.

**Bayesian Reinforcement Learning**    Bayesian RL quantifies the uncertainty or the posterior belief over the MDPs using past experience. Conditioning on the environment's uncertainty, a Bayes-optimal policy can set the optimal balance between exploration and exploitation to maximize the return during training [1, 11, 19, 28]. In a Bayes-adaptive MDP (BAMDP), where the agent augments the state space of MDP with its belief, it is almost impossible to find the optimal policy due to the unknown parameterization and the intractable belief update. VariBAD [56] proposes an approximate but tractable solution that combines meta-learning and approximate variational inference. However, it is restricted to settings where the training and test task distributions are almost the same. Furthermore, the learned latent model is only used as additional information for the policy. LDM uses the learned latent model more actively by creating mixture tasks to train the policy for more general test tasks out of training task distribution.

**Curriculum, Goal and Task Generation**    The idea of generating new tasks for RL is not new. Florensa et al. [10] proposes generative adversarial training to generate goals. Gupta et al. [12] proposes an automatic task design process based on mutual information. SimOpt [3] learns the randomization of simulation parameters based on a few real-world rollouts. POET and enhanced POET [45, 46] generate the terrain for a 2D walker given access to the environment parameters. Dream to Control [14] solves a long-horizon task using a latent-imagination. BIRD [54] learns from

imaginary trajectories by maximizing the mutual information between imaginary and real trajectories. Chandak et al. [2] trains an agent to forecast the future tasks in non-stationary MDPs. Most of these works require prior knowledge or control of the environment parameters, or the pool of the generated tasks is restricted to the training task distribution.

**Data-augmentation for Reinforcement Learning**    Many image augmentation techniques such as random convolution, random shift, l2-regularization, dropout, batch normalization and noise injection are shown to improve the generalization of RL [5, 16, 22, 24, 33, 50]. Mixreg [44] that applies the idea of mix-up [53] in RL, generates new training data as a convex interpolation of input observation and output reward. LDM can be thought of as a data-augmentation method in a way that it generates a mixture task using the data from training tasks. However, LDM generates a new task in the latent space with the latent model that fully encodes the MDPs' dynamics. Instead of the pre-defined augmentation techniques with heuristics, we generate mixture tasks using the learned decoder, which contains the shared structure of the MDPs.

**Out-of-distribution Meta-Reinforcement Learning**    Some recent works aim to generalize RL to OOD tasks [8]. MIER [30] relabels past trajectories in a buffer during a test to generate synthetic training data suited for the test MDP. FLAP [31] learns a shared linear representation of the policy. To adapt to a new task FLAP only needs to predict a set of linear weights for fast adaptation. MetaGenRL [21] meta-learns an RL objective to train a randomly initialized policy on a test task. Most of these methods require experience from the test for additional training or updates for the network. AdMRL [26] performs adversarial virtual training with varying rewards. AdMRL assumes known reward space and parameterization, while LDM meta-learns.

# 5   Experiments

We evaluate LDM and other meta-RL methods on the gridworld example (Figure 1) and three MuJoCo meta-RL tasks [42]. We slightly modify the standard MuJoCo tasks by splitting the task space $\mathcal{M}$ into disjoint $\mathcal{M}_{\text{train}}$ and $\mathcal{M}_{\text{test}}$. We use $\text{RL}^2$ [6] and variBAD [56] as baselines representing the context-based meta-RL methods. LDM without mixture training and the latent dynamics network reduces to $\text{RL}^2$. LDM without mixture training reduces to variBAD if the policy and latent dynamics networks share an encoder. We use E-MAML [40] and ProMP [36] as baselines representing the gradient-based meta-RL methods. We implement $\text{RL}^2$-based Mixreg [44] to evaluate the difference between generating mixture tasks in the latent space and the observation space. Refer to Appendix A for the details of implementations and hyperparameters.

All methods are trained on tasks sampled from $\mathcal{M}_{\text{train}}$ uniformly at random, except for the oracle methods that are trained on tasks in the entire task set $\mathcal{M} = \mathcal{M}_{\text{train}} \cup \mathcal{M}_{\text{test}}$. Note that the oracle performance is only for reference and can not be achieved by any non-oracle methods theoretically. Non-oracle methods without prior knowledge on $\mathcal{M}_{\text{test}}$ require additional exploration during testing to experience the changes from $\mathcal{M}_{\text{train}}$, whereas the oracle agent can exploit the knowledge of the test distribution. In the gridworld example, the oracle agent can search for goals in $\mathcal{M}_{\text{test}}$ before navigating the outermost goal states of $\mathcal{M}_{\text{train}}$. Therefore the main focus should be on the relative improvement of LDM compared to non-oracle methods, bridging the gap between the oracle and the non-oracle methods. We report mean results using 8 random seeds and apply a moving average of window size 5 for all main experiments (4 seeds for ablations). Shaded areas indicate standard deviations for all plots.

## 5.1   Gridworld Navigation

**Experimental setup**    We use the gridworld navigation task introduced in Figure 1. The agent is allowed 5 actions: up, down, left, right, and stay. The reward is 1 for reaching or staying at the hidden goal and $-0.1$ for all other transitions. Each episode lasts for $H = 30$ steps. All baselines are given $N = 4$ rollout episodes for a fixed task except for ProMP and E-MAML that are given $N = 20$ rollouts. Such choice of $N$ follows from the reference implementations of the baselines. The time horizon has been carefully set so that the agent can not visit all states within the first episode but can visit them within two episodes. If a rollout is over, the agent is reset to the origin. The optimal policy is to search for the hidden goal and stay at the goal or return to the goal as quickly as possible. After $N$ episodes, a new task is sampled from $\mathcal{M}_{\text{train}}$ uniformly at random. The reward decoder

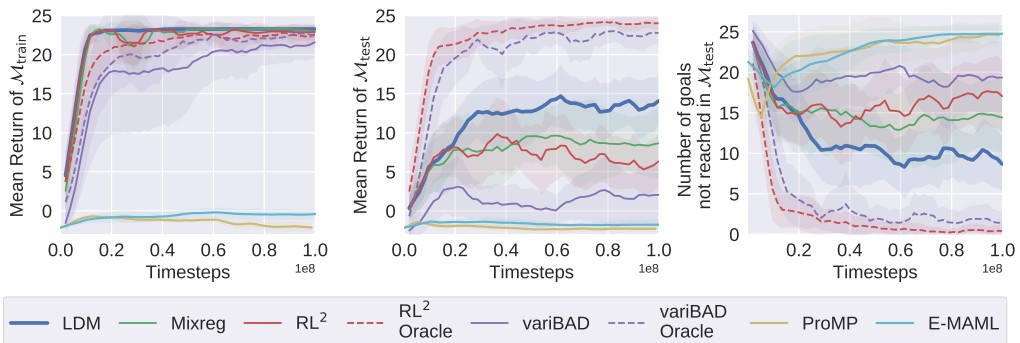

Figure 3: Results for the gridworld task evaluated at the $N$-th episode in terms of the mean returns in $\mathcal{M}_{\text{train}}$ and $\mathcal{M}_{\text{test}}$, and the number of tasks in $\mathcal{M}_{\text{test}}$ in which the agent fails to reach the goal.

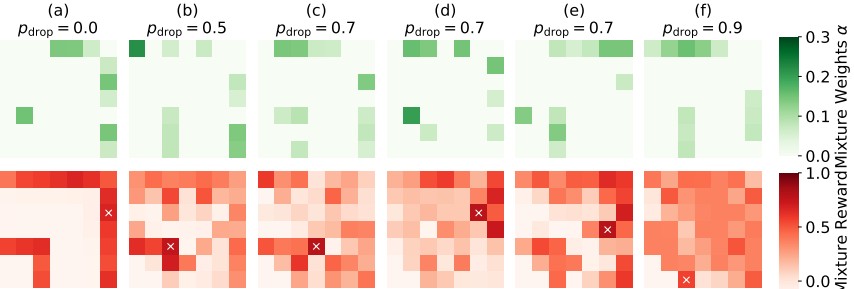

Figure 4: Examples of mixture tasks generated by LDM. **First row**: Mixture weights ($\alpha^{(i)}$) multiplied to the latent models ($m_{H^+}^{(i)}$) at the end of each training task (Equation 1). When the same training tasks are sampled multiple times, we plot the sum of weights. **Second row** (reward map): decoder output for each next state conditioned on the mixture weights from the first row. A cross mark denotes the state with the maximum mixture reward.

requires only the latent model and the next state as input for this task. We apply dropout with the rate $p_{\text{drop}} = 0.7$ to the next state input of the reward decoder.

**Results** Except for ProMP and E-MAML that use feedforward networks, all the baseline methods achieve the same optimal performance in $\mathcal{M}_{\text{train}}$ at the $N$-th episode (Figure 3). However, our method outperforms the non-oracle baselines in $\mathcal{M}_{\text{test}}$. Out of the 27 tasks in $\mathcal{M}_{\text{test}}$, our agent succeeds to visit the goals in 19 tasks on average. Although LDM still does not reach 8 test goals on average, considering that there is no prior information on the test distribution and that the optimal policy for $\mathcal{M}_{\text{train}}$ (dashed path in Figure 1a) does not visit most of the goal states in $\mathcal{M}_{\text{test}}$, the improvement of LDM over RL$^2$ and variBAD is significant. RL$^2$ achieves a high test return initially, but as the policy overfits the training tasks, its test return decays. VariBAD achieves high returns in $\mathcal{M}_{\text{train}}$, but fails to generalize in most of the tasks in $\mathcal{M}_{\text{test}}$. Mixreg performs better than RL$^2$, but does not match LDM.

Refer to Figure 8a for example of LDM's test-time behavior for a task in $\mathcal{M}_{\text{test}}$ that was introduced in Figure 1a. The agent searches for a goal in $\mathcal{M}_{\text{train}}$ first. Once it explores all the goal states in $\mathcal{M}_{\text{train}}$, it starts to search for the goals in $\mathcal{M}_{\text{test}}$. From the second episode, the agent directly heads to the goal based on the context made during the first episode. Note that the initial prior is nonzero even for the goals in $\mathcal{M}_{\text{test}}$ due to the dropout applied to the decoder's state input.

### 5.1.1 Tasks generated by LDM

We present an empirical analysis in Figure 4 to verify that LDM indeed generates meaningful new tasks that help to solve the test tasks. Without prior knowledge or interaction with $\mathcal{M}_{\text{test}}$, it is impossible to create exactly the same reward maps of $\mathcal{M}_{\text{test}}$. But the reward maps in Figure 4c,d,e are sufficient to induce exploration toward the goal states of $\mathcal{M}_{\text{test}}$. Due to the dropout applied to the next state input of the decoder, the decoder assigns high rewards to some goals that belong to

$\mathcal{M}_{\text{test}}$. Note that the decoder does not generalize without dropout therefore assigns high rewards only to training goals (Figure 4a).

### 5.1.2 Extrapolation ability of LDM

To demonstrate that LDM is not restricted to target tasks inside the convex hull of training tasks, we design gridworld-extrapolation task as in Figure 5. This task is similar to the previous gridworld task in Section 5.1, except that we shift the outermost cells of $\mathcal{M}_{\text{train}}$ inside to construct extrapolation tasks $\mathcal{M}_{\text{test2}}$ (Figure 5c). Refer to Figure 6 for the final return for each task when we train LDM ($p_{\text{drop}} = 0.5$) with different values of $\beta$. For small values of $\beta = 1.0$ and $\beta = 1.5$, LDM focuses on training the interpolation tasks in $\mathcal{M}_{\text{test1}}$. As $\beta$ increases, the returns for tasks in $\mathcal{M}_{\text{test1}}$ decrease, but the returns for tasks in $\mathcal{M}_{\text{test2}}$ increase.

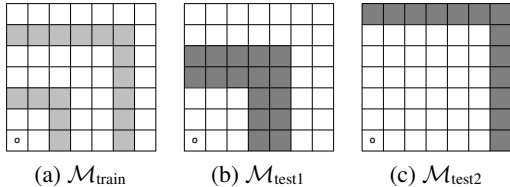

(a) $\mathcal{M}_{\text{train}}$      (b) $\mathcal{M}_{\text{test1}}$      (c) $\mathcal{M}_{\text{test2}}$

Figure 5: Gridworld-extrapolation task.

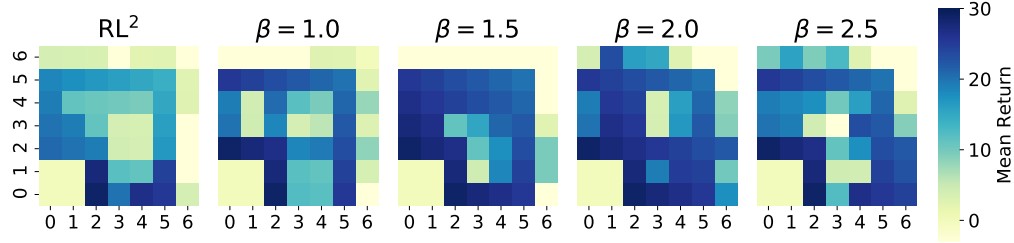

Figure 6: Returns of each task for different extrapolation level $\beta$ on the gridworld-extrapolation task. Mean return for each task in $\mathcal{M}$ at the $N$-th episode, mean of 4 random seeds.

### 5.2 MuJoCo

**Experimental setup** We evaluate our agent and baselines on three standard MuJoCo domains (Ant-direction, Ant-goal, and Half-cheetah-velocity) to verify how effective LDM is for continuous control tasks. We evaluate the agents on a fixed set of evaluation tasks $\mathcal{M}_{\text{eval}} \subset \mathcal{M}_{\text{test}}$ for each domain to ensure the test result is not affected by the sampling of evaluation tasks (Table 1). For MuJoCo tasks we report the results of PEARL [35], MQL [8], and MIER [30] with number of rollouts per iteration ($N$) equal to 3, 11 and 3 respectively. ProMP and E-MAML are given $N = 20$ and all other methods are given $N = 2$ rollouts (following the reference implementations). $H = 200$ steps for each episode of all MuJoCo tasks. Because PEARL, MQL, and MIER use an off-policy RL algorithm, their performance converges with much less training data. Therefore, for the three baselines, we report the converged asymptotic performance after 5 million steps of environment interaction. We set LDM's $p_{\text{drop}} = 0.5$ for all MuJoCo tasks.

**Ant-direction** We construct the Ant-direction task based on the example discussed in the introduction. $\theta$ denotes the angle of the target direction from the origin. A major component of the reward is the dot product of the target direction and the agent's velocity.

**Ant-goal** We construct the Ant-goal task similar to the gridworld task, where the agent has to reach the unknown goal position. The training task distribution $\mathcal{M}_{\text{train}}$ (shaded region of Figure 8c) is continuous, unlike the gridworld and Ant-direction tasks. $r$ and $\theta$ denote the radius and angle of the

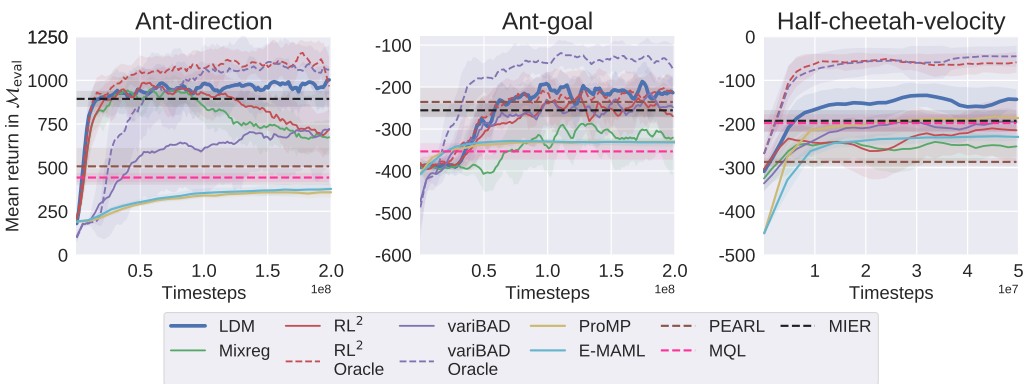

Figure 7: Mean returns at the $N$-th episodes in $\mathcal{M}_{\text{eval}}$ of three MuJoCo tasks.

Table 1: Set of training, test and evaluation tasks of Mujoco tasks. $k \in \{0, 1, 2, 3\}$.

| | Ant-direction $\theta$ | Ant-goal $r$ | $\theta$ | Half-cheetah-velocity $v$ |
|---|---|---|---|---|
| $\mathcal{M}_{\text{train}}$ | $90° \times k$ | $[0.0, 1.0) \cup [2.5, 3.0)$ | $[0°, 360°)$ | $[0.0, 0.5) \cup [3.0, 3.5)$ |
| $\mathcal{M}_{\text{test}}$ | $90° \times k + 45°$ | $[1.0, 2.5)$ | $[0°, 360°)$ | $[0.5, 3.0)$ |
| $\mathcal{M}_{\text{eval}}$ | $90° \times k + 45°$ | $1.75$ | $90° \times k$ | $\{0.75, 1.25, 1.75, 2.25, 2.75\}$ |

goal from the origin, respectively. A major component of the reward is the negative value of the taxicab distance between the agent and the goal. We set $\beta = 2$ because LDM with $\beta = 1$ generates goals near the origin mostly due to the symmetry of training tasks. Refer to Appendix C for a detailed analysis regarding the choice of $\beta$. One may argue that the training and test tasks are not disjoint because the agent passes through the goal states of $\mathcal{M}_{\text{test}}$ momentarily while learning to solve $\mathcal{M}_{\text{train}}$. However, the task inference with the unseen rewards from $\mathcal{M}_{\text{test}}$ cannot be learned during training. The policy to stay at the goals of $\mathcal{M}_{\text{test}}$ until the end of the time horizon is also not learned.

**Half-cheetah-velocity**  We train the Half-cheetah agent to match the target velocity sampled from the two distributions at extremes and test for target velocities in between. This task is relatively easier than the previous Ant tasks due to the reduced dimension of the task space. Therefore we train more mixture workers than the other tasks, training $n = 12$ normal workers and $\hat{n} = 4$ mixture workers in parallel. Refer to Appendix D for additional results with different numbers of mixture workers. The target velocity is $v$, where the velocity is measured as a change of position per second (or 20 environment steps). A major component of the reward is the negative value of the difference between the target velocity and the agent's velocity. Similar to Ant-Goal, although the test velocities in $\mathcal{M}_{\text{test}}$ are achieved momentarily in the process of reaching a target velocity in $[3.0, 3.5)$, the agent does not learn to maintain the target velocities in $\mathcal{M}_{\text{test}}$ during training in $\mathcal{M}_{\text{train}}$.

**Results**  Refer to Figure 7 for the test returns in $\mathcal{M}_{\text{eval}}$. LDM outperforms the non-oracle baselines at the $N$-th episode. VariBAD oracle achieves the best test returns for all MuJoCo tests, revealing the strength of variBAD's task inference ability for tasks seen during training. However, the non-oracle variBAD has difficulty generalizing to $\mathcal{M}_{\text{test}}$. The performance of Mixreg is lower than that of $RL^2$, which reveals the limitation of mixing in complex observation space. In all MuJoCo tasks, the agent can infer the true task dynamics using the reward at every timestep, even at the first step of an episode. Therefore gradient-based methods with feedforward networks also make progress, unlike the gridworld task. Because PEARL is designed to collect data for the first two episodes, it achieves low returns before the $N$-th episode. Even after accumulating context, the policy of PEARL is not prepared for the unseen contexts in Ant-direction and Half-cheetah-velocity. Note that MQL and MIER require additional buffer-based re-labeling or re-training for testing after collecting some rollouts of the test task. LDM can prepare in advance during training so that we do not require collection of test rollouts and extra buffer-based training during testing.

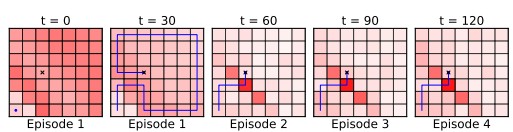
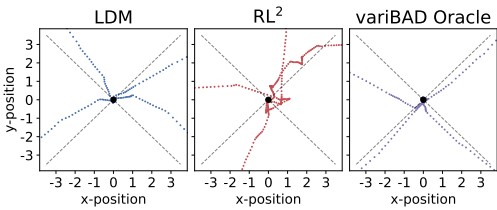
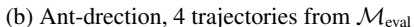

(a) Sample trajectory of LDM for 4 rollout episodes. The red shade denotes decoder output for each state conditioned on the online context.

(b) Ant-drection, 4 trajectories from $\mathcal{M}_{\text{eval}}$.

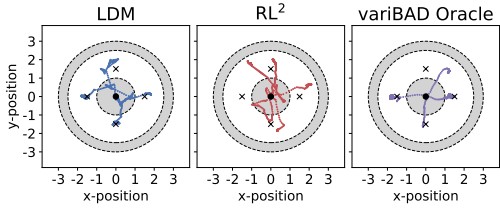
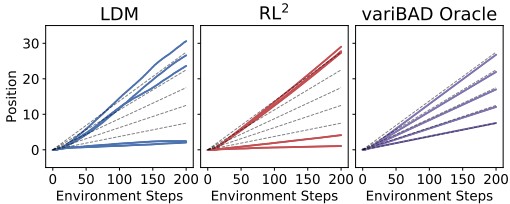

(c) Ant-goal, 4 trajectories from $\mathcal{M}_{\text{eval}}$.

(d) Half-cheetah-velocity, 5 trajectories from $\mathcal{M}_{\text{eval}}$.

Figure 8: Example trajectories of the agents in $\mathcal{M}_{\text{eval}}$. We illustrate the behavior at the $N$-th episode as colored paths. The targets of $\mathcal{M}_{\text{eval}}$ are indicated as dashed lines or cross marks

Because LDM is not trained in $\mathcal{M}_{\text{test}}$, it can not solve the target tasks optimally from the beginning of each task in the Ant-direction and Ant-goal tasks, unlike variBAD Oracle (Figure 8b and 8c). However, LDM reaches the target direction or goal close enough after a sufficiently small amount of exploration, unlike RL$^2$. For the Half-cheetah-velocity task, LDM generates well-separated policies compared to RL$^2$ (Figure 8d). Refer to Appendix F for additional experimental results on the test returns at the first rollout episode, the training returns, and sample trajectories on training tasks.

## 6 Conclusion

We propose Latent Dynamics Mixture to improve the generalization ability of meta-RL by training policy with generated mixture tasks. Our method outperforms baseline meta-RL methods in experiments with strictly disjoint training and test task distributions, even reaching the oracle performance in some tasks. Because our latent dynamics network and the task generation process are independent of the policy network, we expect LDM to make orthogonal contributions when combined with not only RL$^2$ but most of other meta-RL methods.

We believe that our work can be a starting point for many interesting future works. For example, instead of a heuristic weight-sampling for generating the mixture, we may incorporate OOD generation techniques in the latent space. Another extension is to train the latent dynamics network to decode not only the reward but also the state transition dynamics. Then we can generate a purely imaginary mixture task without additional environment interaction.

## Acknowledgments

This work was supported by the National Research Foundation of Korea (NRF) grant funded by the Korea government (MSIT) [2021R1A2C2007518]. We thank Dong Hoon Lee, Minguk Jang, and Dong Geun Shin for constructive feedback and discussions.

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
