# A  Implementation Details

## A.1  Baselines

**$RL^2$ and variBAD**  We use the open-source reference implementation of variBAD at `https://github.com/lmzintgraf/varibad` to report the results of $RL^2$ and variBAD. All gridworld and MuJoCo environments used for our experiments are based on this implementation. We modify the environments to contain separate $\mathcal{M}_{\text{train}}$ and $\mathcal{M}_{\text{test}}$. The oracle versions of $RL^2$ and variBAD use the original environment $\mathcal{M}$. We keep all the network structures and hyperparameters of the reference implementation except for the gridworld task. We increase the GRU hidden size from 64 to 128 and the GRU output size from 5 to 32 for the gridworld task because our gridworld task ($7 \times 7$) has more cells than the gridworld task in variBAD ($5 \times 5$).

**LDM**  We implement LDM based on the implementation of variBAD. Refer to Table 3 for the hyperparameters used to train LDM. Most of the network structures and hyperparameters follow the reference implementation of variBAD. The policy network of LDM is from the $RL^2$ implementation of variBAD. The latent dynamics network is from the VAE part of variBAD. VariBAD uses a multi-head structure (each head for a goal state in $\mathcal{M}_{\text{train}}$) with binary cross-entropy (BCE) loss for the decoder output for the gridworld task. Because LDM needs to generate rewards for tasks out of $\mathcal{M}_{\text{train}}$ as well, we modify the decoder to a general single head structure. The latent model $m$ is multidimensional, therefore we sample the weight corresponding to each dimension independently. The weights in 4a represent the mean weights of all dimensions. Refer to our reference implementation at `https://github.com/suyoung-lee/LDM`.

**Mixreg**  The original Mixreg is based on Deep Q-Network (DQN). Therefore, we implement a variant of Mixreg based on $RL^2$ by modifying the code we use for $RL^2$. We use the same Dirichlet mixture weights used for LDM and multiply the weights to the states and rewards to generate mixture tasks. We keep the ratio between the true tasks and mixture tasks as 14:2.

**ProMP and E-MAML**  We use the open-source reference implementation of ProMP at `https://github.com/jonasrothfuss/ProMP` for ProMP and E-MAML. We only modify the environments to contain separate $\mathcal{M}_{\text{train}}$ and $\mathcal{M}_{\text{test}}$ and keep all the reference implementation setup.

**PEARL, MQL, and MIER**  We use the open-source reference implementation of PEARL at `https://github.com/katerakelly/oyster`, MQL at `https://github.com/amazon-research/meta-q-learning` and MIER at `https://github.com/russellmendonca/mier_public`. We only modify the environments to contain separate $\mathcal{M}_{\text{train}}$ and $\mathcal{M}_{\text{test}}$ and keep all the reference implementation setup. We report the performance at 5 million steps as asymptotic performance for MuJoCo tasks.

## A.2  Runtime Comparison

We report the average runtime spent to train Half-cheetah-velocity for 5e7 environment steps (5e6 steps for PEARL) in Table 2. We ran multiple experiments on our machine (Nvidia TITAN X) simultaneously, therefore consider the following as relative ordering of complexity.

Table 2: Mean runtime to train Half-cheetah-velocity.

|  | LDM | Mixreg | $RL^2$ | variBAD | ProMP | E-MAML | PEARL |
|---|---|---|---|---|---|---|---|
| Half-cheetah-velocity Runtime (hours) | 31 | 28 | 25 | 10 | 2 | 2 | 25 |

ProMP and E-MAML require the least training time because they do not use recurrent networks. VariBAD requires less train time than $RL^2$ because variBAD does not backpropagate the policy network's gradient to the recurrent encoder. LDM requires more training time than $RL^2$ because LDM trains the policy network and a separate latent dynamics network.

## A.3 Hyperparameters

Table 3: Hyperparameters of LDM.

| | | | Gridworld | MuJoCo |
|---|---|---|---|---|
| RL algorithm | | | A2C
Epsilon: 1.0e-5
Discount: 0.95
Max grad norm: 0.5
Value loss coeff.: 0.5
Entropy coeff.: 0.01
GAE parameter tau: 0.95 | PPO
Batch size: 3200
Minibatches: 4
Max grad norm: 0.5
Clip parameter: 0.1
Value loss coeff.: 0.5
Entropy coeff.: 0.01 |
| Number of steps of a rollout episode ($H$) | | | 30 | 200 |
| Number of rollout episodes per iteration ($N$) | | | 4 | 2 |
| Extrapolation level ($\beta$) | | | 1.0 | 1.0 (2.0 for Ant-goal) |
| Decoder input dropout rate ($p_{\text{drop}}$) | | | 0.7 | 0.5 |
| Number of parallel processes | Normal workers ($n$) | | 14 | 14 (12 for Cheetah-vel) |
| | Mixture workers ($\hat{n}$) | | 2 | 2 (4 for Cheetah-vel) |
| Policy Network | Encoder-p ($\phi_p$) | State encoding | 1 FC layer, 32 dim | 1 FC layer, 32 dim |
| | | Action encoding | 0 dim | 1 FC layer, 16 dim |
| | | Reward encoding | 1 FC layer, 8 dim | 1 FC layer, 16 dim |
| | | GRU | 128 hidden size | 128 hidden size |
| | | GRU output ($b_t$) size | 32 | 128 |
| | Policy ($\pi_\psi$) | | 1 FC layer, 32 nodes | 1 FC layer, 128 nodes |
| | Activation | | tanh | tanh |
| | Learning rate | | 7.0e-4 | 7.0e-4 |
| Latent Dynamics Network | Encoder-v ($\phi_v$) | State encoding | 1 FC layer, 32 dim | 1 FC layer, 32 dim |
| | | Action encoding | 0 dim | 1 FC layer, 16 dim |
| | | Reward encoding | 1 FC layer, 8 dim | 1 FC layer, 16 dim |
| | | GRU | 128 hidden size | 128 hidden size |
| | | Task embedding ($m_t$) size (sample from GRU output dimension) | 32 | 5 for Ant-direction and Cheetah-vel
10 for Ant-goal |
| | Reward decoder ($\theta_R$) | | 2 FC layers, 32 and 32 nodes | 2 FC layers, 64 and 32 nodes |
| | Decoder loss function | | BCE | MSE |
| | Activation | | ReLU | ReLU |
| | Learning rate | | 0.001 | 0.001 |
| | Buffer size | | 100000 | 10000 |

# B  Test-time Generalization of the Latent Model

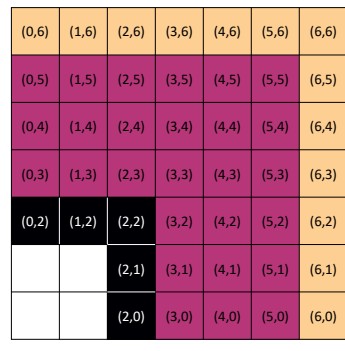

(a) Gridworld: sample tasks.

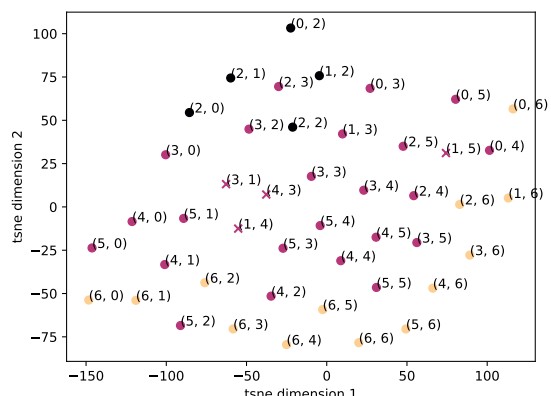

(b) Gridworld: t-SNE plot of test-time latent model $m_{H+}$. The cross marks denote the tasks that the LDM agent fails to visit.

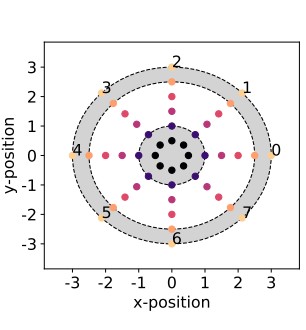

(c) Ant-goal: sample tasks.

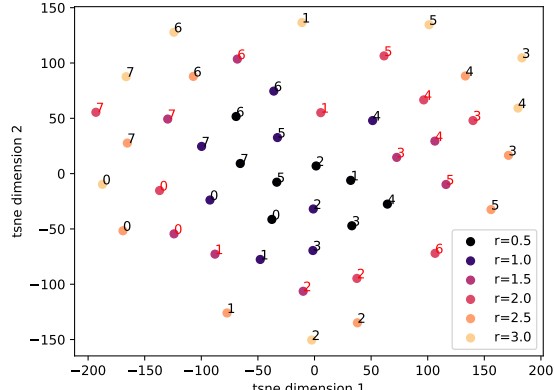

(d) Ant-goal: t-SNE plot of test-time latent model $m_{H+}$. The number on each point times $45°$ denotes the target goal's angle from the origin. Latent models with red numbers belong to $\mathcal{M}_{\text{test}}$.

Figure 9: Latent dynamics network's learned latent models on 45 tasks in the gridworld and 48 tasks in Ant-goal.

We empirically demonstrate that the structure of the test task is well reflected in the latent models although the latent dynamics network is not trained in $\mathcal{M}_{\text{test}}$ and $\hat{M}$ (Figure 9). For each task in the gridworld task, we collect the latent model at the last step ($t = H^+$). Then we reduce the dimension of the collected latent models to two dimensions via t-SNE (Figure 9b). The latent models of the tasks in $\mathcal{M}_{\text{test}}$ are between the inner subset and the outer subset of the training tasks. Similarly, we evaluate the latent dynamics model in Ant-goal. We evaluate the latent models on 48 tasks as in Figure 9c where $r \in \{0.5, 1.0, 1.5, 2.0, 2.5, 3.0\}$ and $\theta \in \{0°, 45°, 90°, 135°, 180°, 225°, 270°, 315°\}$. Although LDM is not trained on the tasks with $r = 1.5$ and $r = 2.0$, the latent models are between the training tasks' latent models (Figure 9d). On the other hand, the policy network can not be trained stably with a large input dropout (RL$^2$ dropout in Figure 13). These empirical results support our claim that our latent dynamics network with dropout can generalize to unseen test dynamics although the policy can not. Therefore the mixtures of the latent models can generate tasks similar to the test tasks.

## C  Analysis on the Ant-goal Task (Extrapolation Level $\beta$)

At the beginning of an iteration, we sample $n = 14$ training tasks from $\mathcal{M}_{\text{train}}$ on the Ant-goal task. Because we sample a sufficiently large number of training tasks, a mixture task's goal is located near the origin with high probability if we set $\beta = 1.0$ (Figure 10). Therefore we use $\beta = 2.0$ that effectively improves the test returns on the Ant-goal task.

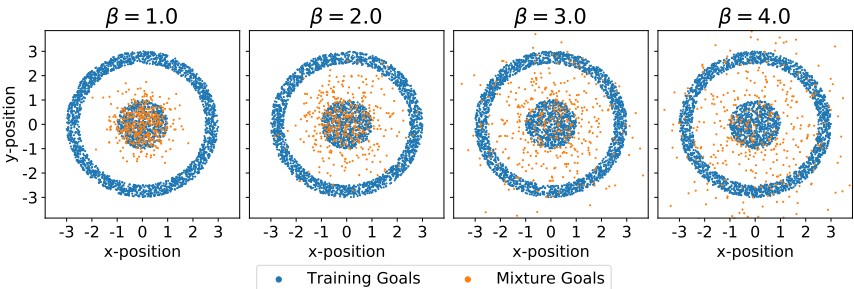

Figure 10: Expected mixture goals for different extrapolation level $\beta$ on the Ant-goal task. We sample training tasks from $\mathcal{M}_{\text{train}}$ for 200 iterations. For each iteration, we sample $n = 14$ goals for normal workers. Then we mix the goal coordinates of the training tasks using the Dirichlet weights to generate $\hat{n} = 2$ mixture goals. We plot the coordinates of the training and mixture goals.

## D  Number of Mixture Workers $\hat{n}$

Refer to Figure 11 for the returns in $\mathcal{M}_{\text{eval}}$ and $\mathcal{M}_{\text{eval-train}}$ (Table 4) on the Half-cheetah-velocity task for different values of $\hat{n}$. We report the results for $\hat{n} \in \{2, 3, 4, 5\}$ and keep the total number of workers fixed at 16. For all values of $\hat{n}$, LDM outperforms RL$^2$ in test returns at the beginning of training. However, the test return of LDM decays as training tasks dominate the policy updates. If $\hat{n}$ is small, training of the policy is easily dominated by the training tasks, and the test return converges that of RL$^2$ quickly. If $\hat{n}$ is too large, normal workers have difficulty in learning the optimal policy for training tasks ($\hat{n} = 5$ in Figure 11, second row). We use $\hat{n} = 4$ that sets a balance in between.

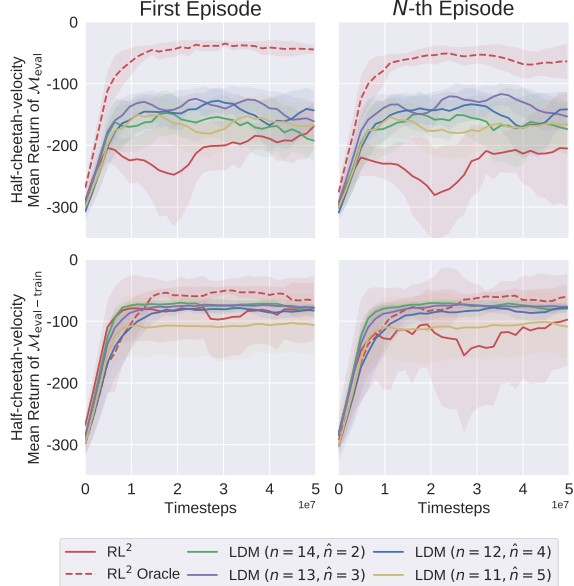

Figure 11: Training and test returns for different numbers of mixture workers $\hat{n}$ on Half-cheetah-velocity, using 4 random seeds.

# E Ablations on Dropout

## E.1 Amount of Dropout $p_{\text{drop}}$

We report the performance of LDM with different dropout rates $p_{\text{drop}} \in \{0.0, 0.5, 0.7, 0.9\}$ in Figure 12. LDM without the input dropout ($p_{\text{drop}} = 0.0$) slightly outperforms RL$^2$ at the end of training, but the improvement is insignificant. The test performance improves as the dropout rate increases. But if the rate is too large ($p_{\text{drop}} = 0.9$), it becomes difficult to train the decoder and the performance decreases.

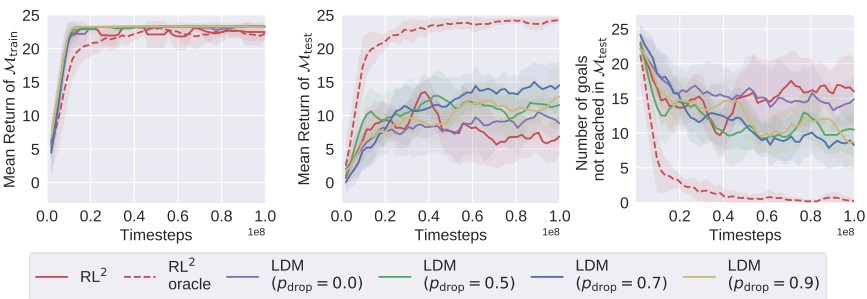

Figure 12: Results of LDM with different dropout rate on gridworld task. Evaluated at the $N$-th episode, using 4 random seeds.

## E.2 Dropout on other baselines

To demonstrate that dropout is not all the regularization required to achieve better generalization, we evaluate RL$^2$ dropout and variBAD dropout, both with $p_{\text{drop}} = 0.7$ in Figure 13. RL$^2$ doesn't use a decoder, therefore we apply dropout on the state input of the policy encoder. RL$^2$ dropout cannot be trained stably, since training policy network with a multi-step policy gradient loss is much more complex than training the decoder with a single-step regression loss. For variBAD dropout, we apply dropout on the state input of the decoder (same as LDM). VariBAD shares an encoder for both VAE and the policy network and does not backpropagate the policy loss to the encoder. Therefore, the encoding, trained with decoder with dropout, may disturb the policy network from stable training. We evaluate LDM with a shared encoder, where we use a single encoder instead of separate encoder-p and encoder-v. We encountered instability in policy training when the encoding, trained for a decoder with dropout, was used for policy. If we have separate encoders, even when the VAE is not perfectly trained due to the dropout, it does not affect the policy.

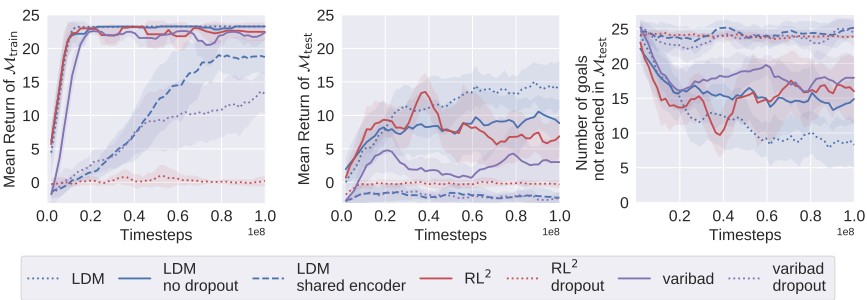

Figure 13: Results of LDM, RL$^2$, and variBAD trained with and without dropout on gridworld task. Evaluated at the $N$-th episode, using 4 random seeds.

# F    Additional Experimental Results

## F.1    Gridworld First Rollout Episode

The non-oracle methods spend most of the timesteps in the first episode to explore the goal states of $\mathcal{M}_{\text{train}}$. Therefore the mean returns in $\mathcal{M}_{\text{test}}$ are significantly lower than those at the $N$-th episode (Figure 3). The oracle methods explore the goal states of $\mathcal{M}_{\text{test}}$ before exploring the outermost states of $\mathcal{M}_{\text{train}}$ during the first episode. Therefore the mean returns of oracle methods in $\mathcal{M}_{\text{train}}$ are lower than those of non-oracle methods.

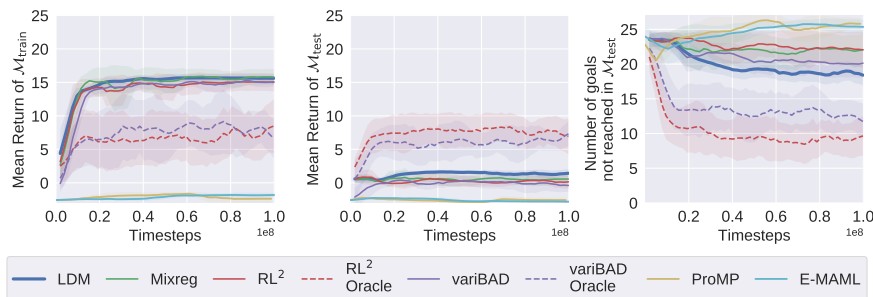

Figure 14: Results of the gridworld task evaluated at the first episode in terms of the mean returns in $\mathcal{M}_{\text{train}}$ and $\mathcal{M}_{\text{test}}$, and the number of tasks in $\mathcal{M}_{\text{test}}$ in which the agent fails to reach the goal.

## F.2    MuJoCo First Rollout Episode

Since the task can be inferred from the reward at any timestep of training, the performance of LDM at the first episode is nearly the same as that at the $N$-th episode (Figure 7). ProMP, E-MAML, and PEARL need to collect trajectories until the $N$-th episode.

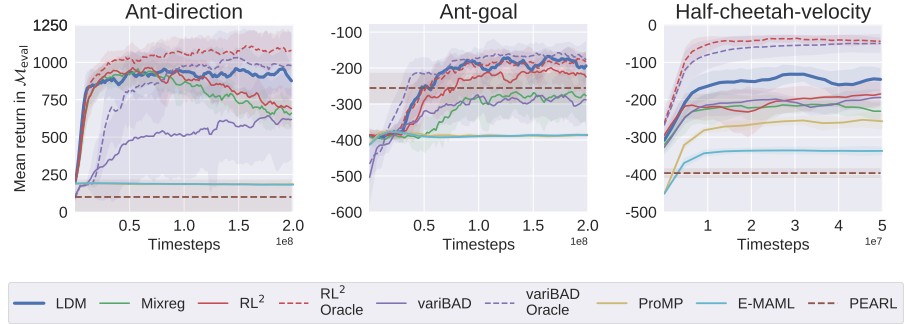

Figure 15: Mean returns at the first episodes in $\mathcal{M}_{\text{eval}}$ of three MuJoCo tasks.

## F.3    MuJoCo Training Results

As we defined $\mathcal{M}_{\text{eval}} \subset \mathcal{M}_{\text{test}}$, we define $\mathcal{M}_{\text{eval-train}} \subset \mathcal{M}_{\text{train}}$ to report the training results (Table 4).

Table 4: $\mathcal{M}_{\text{eval-train}}$ for MuJoCo tasks. $k \in \{0, 1, 2, 3\}$.

|  | Ant-direction | Ant-goal | | Half-cheetah-velocity |
|---|---|---|---|---|
|  | $\theta$ | $r$ | $\theta$ | $v$ |
| $\mathcal{M}_{\text{eval-train}}$ | $90° \times k$ | $\{0.50, 2.75\}$ | $90° \times k$ | $\{0.25, 3.25\}$ |

Refer to Figure 16 for the mean returns on the training tasks and Figure 17 for the trajectories. LDM achieves higher training returns for all tasks than its baseline $RL^2$, although LDM devotes some

portion of training steps to train mixture tasks. LDM achieves the best training performance in the Ant-direction task while RL[2]'s performance gradually decreases. When there are only a few training tasks, RL[2]-based methods often collapse into a single-mode, unable to construct sharp decision boundaries between tasks (Figure 17a). VariBAD achieves high training returns in Ant-direction and Ant-goal, although its test returns in $\mathcal{M}_{\text{test}}$ are low.

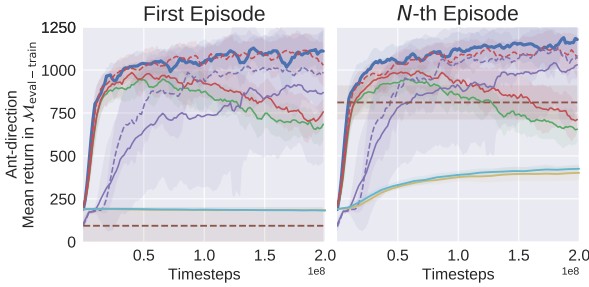

(a) Ant-direction, mean return of the 4 tasks in $\mathcal{M}_{\text{eval-train}}$

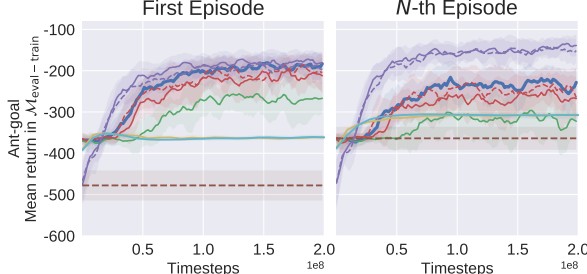

(b) Ant-goal, mean return of the 8 tasks in $\mathcal{M}_{\text{eval-train}}$

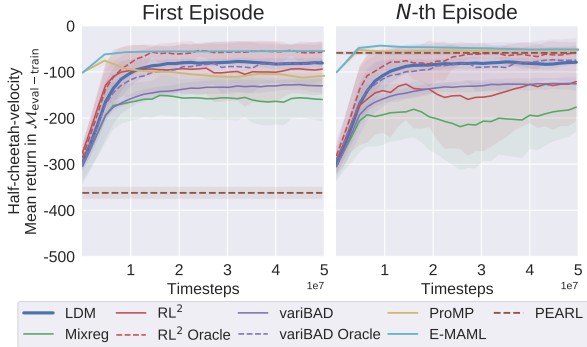

(c) Half-cheetah-velocity, mean return of the 2 tasks in $\mathcal{M}_{\text{eval-train}}$

Figure 16: Mean return in $\mathcal{M}_{\text{eval-train}}$ of three MuJoCo tasks.

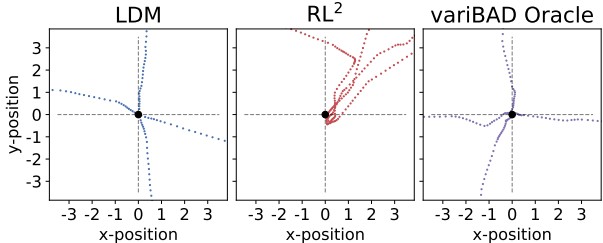

(a) Ant-drection, 4 trajectories from $\mathcal{M}_{\text{eval-train}}$ in each plot. The target directions are dashed lines.

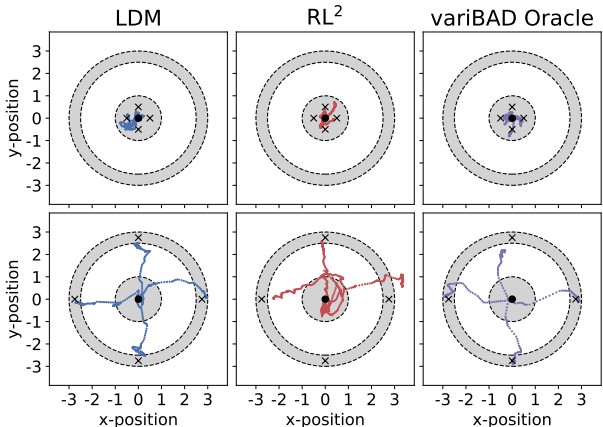

(b) Ant-goal, 8 trajectories from $\mathcal{M}_{\text{eval-train}}$ for each method (4 in each plot). The cross marks are the goal positions.

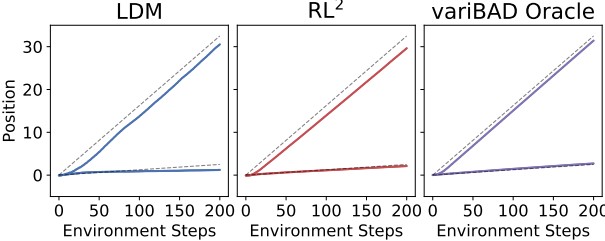

(c) Half-cheetah-velocity, 2 trajectories from $\mathcal{M}_{\text{eval-train}}$ in each plot.

Figure 17: Example trajectories of the agents in $\mathcal{M}_{\text{eval-train}}$ of MuJoCo tasks. We illustrate the behavior at the $N$-th episode as colored paths. The targets of $\mathcal{M}_{\text{eval-train}}$ are indicated as dashed lines or cross marks