# OpenReview forum: "Improving Generalization in Meta-RL with Imaginary Tasks from Latent Dynamics Mixture"
_NeurIPS.cc/2021/Conference — NeurIPS 2021 Poster_

### Official Review · Reviewer_BvaU · 2021-07-16

**Rating:** 5
**Confidence:** 4

**Summary:**

This paper focuses on the field of meta-reinforcement learning (meta-RL), where the authors propose a method to augment the tasks for meta-reinforcement by applying a mixture of learned reward functions. Specifically, the paper builds on the framework of variBad [1], which uses a VAE to learn a latent dynamics and reward model by encoding the data and decoding the next states and rewards. The authors propose a method that uses a mixture of the encoded latent variables from different tasks during the training time to obtain new latent encodings corresponding to imaginary tasks, and then applies these latent encodings to the reward decoder to obtain new reward functions to train the policy on these imaginary tasks.

The authors then evaluate the proposed method on grid world tasks and several MuJoCo locomotion tasks, where the authors specifically emphasize the performance of adapting to out-of-distribution tasks. The empirical results demonstrate that the proposed method outperforms several baselines.


References

[1] Zintgraf, L., et al. "VariBAD: a very good method for Bayes-adaptive deep RL via meta-learning." Proceedings of ICLR 2020 (2020).


**Limitations And Societal Impact:**

See main review for limitations and N/A for social impact.

**Main Review:**

Overall I think this paper presents an interesting idea on task augmentation for metal-reinforcement learning. The paper is well-written and the authors include a lot of experiment details and ablation studies in the paper. However, there are also many limitations in this paper that need to be addressed.

Pros:

The main idea is simple and intuitive. Using a mixture of latent codes to augment the task is a natural idea and it is easy to believe that it should work well.

The paper is well written. The main idea is presented in a way that is easy to understand, and the whole paper is structured in an organized way.

The authors provide detailed analysis of the experiments, with many informative behavior visualizations and ablation studies about the proposed method.

Cons:

Several important baseline comparisons are missing. The main advantage of the proposed method is the capability of adapting well to out-of-distribution tasks, where the training tasks and test tasks come from non-overlapping distributions. As the authors mentioned in related works, several prior works such as MQL [1], MIER [2] and FLAP [3] have attempted to address the out-of-distribution problem of test time tasks. Since this paper addresses the same problem, it would be important to compare the proposed method to these baselines.

The scope of experiments in this paper is rather limited. Besides simple grid world environments, the authors only include 3 MuJoCo locomotion environments, which I don’t believe is enough. In order to add more environments, the authors could consider out-of-distribution environments proposed in [1, 2], and also MetaWorld [4].

In summary, being an empirical paper, this paper does not include enough experiments to convince me about the performance of the proposed method. Therefore, it is hard to recommend acceptance of this paper at the current state.


References

[1] Fakoor, Rasool, et al. "Meta-q-learning." arXiv preprint arXiv:1910.00125 (2019).

[2] Mendonca, Russell, et al. "Meta-reinforcement learning robust to distributional shift via model identification and experience relabeling." arXiv preprint arXiv:2006.07178 (2020).

[3] Peng, Matt, Banghua Zhu, and Jiantao Jiao. "Linear representation meta-reinforcement learning for instant adaptation." arXiv preprint arXiv:2101.04750 (2021).

[4] Yu, Tianhe, et al. "Meta-world: A benchmark and evaluation for multi-task and meta reinforcement learning." Conference on Robot Learning. PMLR, 2020.


**Time Spent Reviewing:**

4

---

> ### Author Response · Authors · 2021-08-10
> **Author Response to Reviewer BvaU**
>
> Thank you for your interest in our main ideas and for your positive comments on the intuition of our method. In order to fully convince the reviewer about the performance of our method, we provide more experimental results as following.
>
> **Q1) Several important baselines that address OOD problems are missing**
>
> The main reason we did not compare these methods ([1],[2], and [3]) is because we thought they were dealing with a different problem setting. As the reviewer pointed, the main advantage of our method is the "capability of adapting well to out-of-distribution tasks" but "without test-time training". These off-policy methods have an advantage in training sample efficiency, but they require additional extensive re-labeling or re-training for testing (using buffer of size 1e6 steps collected during meta training) after collecting some rollouts of the test task. The purpose of our method is to prepare in advance during training so that we do not require collection of test rollouts and extra buffer-based training during testing. Note that LDM can achieve high returns even at the very first rollout episode in MuJoCo tasks (Figure 14 of Appendix F.2).
>
> For our experiment, we compared LDM with methods that do not require additional training in tests such as RL^2, variBAD, and mixreg, or methods that require only a small number of updates such as MAML, ProMP, and PEARL without using buffer collected during meta-training. We apologize for the lack of explanation in the experiment section about the criteria for selecting the baselines.
>
> Although we think the conditions for solving the problem are different, therefore it is difficult to make a fair comparison, they share a common goal to solve OOD tasks as Reviewer BvaU pointed. Therefore, we run experiments for MQL and MIER on MuJoCo tasks, the baselines that provide public reference implementations. We use the reference implementations and only modify the environment as our setting. These methods are highly sample-efficient compared to on-policy methods, requiring only 1M~5M steps to converge. Similar to how we report the score for PEARL in Figure 5, we report the asymptotic test return at 5M steps. We report the result in Table R5 and Figure R3 below.
>
> **Table R5. Comparison to MQL and MIER, test return at the N-th episode**
>
> |                  |      | mean     | std     | 95% conf interval |
> |------------------|------|----------|---------|-------------------|
> |   Ant-direction  | LDM  | 1002.510 | 78.633 | 54.490            |
> |                  | MQL  | 444.328  | 26.232  | 18.178            |
> |                  | MIER | 893.876  | 45.589  | 31.592            |
> |     Ant-goal     | LDM  | -213.593 | 36.213  | 25.094            |
> |                  | MQL  | -353.330 | 19.546  | 13.545            |
> |                  | MIER | -255.415 | 16.318  | 11.308            |
> | Cheetah-velocity | LDM  | -143.682 | 23.378  | 16.200            |
> |                  | MQL  | -198.099 | 20.242  | 14.027            |
> |                  | MIER | -192.678 | 24.452  | 16.944            |
>
> **Link to Figure R3. MuJoCo Test returns including MQL and MIER)**
>
> https://drive.google.com/file/d/1-MspniJ_EwJ57Raviu_diYngVS5I6uxn/view?usp=sharing
>
> This figure corresponds to Figure 5 of the main manuscript. Currently MQL and MIER are in 8 seeds, the rest are in 4 seeds. This graph will be updated as soon as we as soon as we obtain the remaining results with 8 seeds.
>
> **(updated on August 17th)** We have completed additional 4 runs for the remaining baselines on MuJoCo tasks. We report the final result in our common comment for all reviewers.
>
> The main focus of MQL is on standard meta-learning setting with i.i.d. task sampling, since it only evaluates on one out-of-distribution task (half-cheetal-vel-ood task in Appendix). In Figure R3, MQL outperforms most of baselines on Half-cheetah-velocity task but does not work well on more complex ant tasks. On the other hand, MIER is a work that specifically focuses on OOD tasks, outperforming MQL in all environment. LDM consistently outperforms MIER on the MuJoCo tasks we evaluated.
>
> **Q2) Add more environments**
>
> As Reviewer BvaU suggested, we present additional experimental results on OOD Humanoid-direction task in Table R6 and Figure R4. We create the task space the same as the Ant-directions task in Table 1 of the main manuscript.
> Since we evaluate on disjoint training and test settings with only varying reward dynamics, there were restrictions on the pool of environments we could use (e.g., cannot use forward-backward tasks, tasks with varying state-transition parameters). Also, there is a restriction arising from that we tried to use the reference implementations of baselines without changing any hyperparameters and network structures but only splitting the training and test environment. Humanoid-direction task was the only remaining task free from these restrictions. Most related work runs experiments on 4~6 MuJoCo tasks including tasks that are restricted for our experiment: variBAD(4), PEARL(6), proMP(6), MQL(6), MIER(5)).
>
> **Table R6. Humanoid-direction Task, test return at the N-th episode**
>
> |                | mean     | std      | 95% conf. interval |
> |----------------|----------|----------|--------------------|
> | LDM            | 984.522  | 84.160    | 58.320             |
> | Mixreg         | 392.701  | 182.186  | 126.248            |
> | RL2            | 290.435  | 56.909   | 39.436             |
> | RL2 oracle     | 1020.316 | 136.565  | 94.635             |
> | variBAD        | 292.178  | 102.924  | 71.323             |
> | variBAD oracle | 1206.785 | 97.778   | 67.757             |
> | proMP          | 370.046  | 24.372   | 16.889             |
> | E-MAML         | 338.147  | 33.644   | 23.314             |
> | PEARL          | 542.415  | 84.873| 58.814             |
> | MQL            | 904.417 | 13.816 | 9.574              |
> | MIER           | 936.819 | 41.399 | 28.688             |
>
> **Link to Figure R4. MuJoCo Humanoid-direction Test Returns)**
>
> https://drive.google.com/file/d/1ChJljeQkTCJKe6S6IfxpkcaKg7nOBfHA/view?usp=sharing
>
> It is convincing that the performance of LDM converges to the performance of RL^2 oracle similar to the ant-direction result.
> In addition, although the two gridworld tasks we use (one in the main manuscript, one in Appendix B.2.) seem to be simple, we think that they are more complex than the MuJoCo tasks in terms of exploration required to find unseen test goals. Unlike MuJoCo, where the reward signal is given at any step to allow inference on the task information, the sparse reward in gridworld is only given at the unknown rewarding state. Therefore, we think that the gridworld task is the task that can best evaluate the exploration ability required to infer the task information and adapt to an unseen environment.

---

> > ### Comment · Reviewer_BvaU · 2021-08-18
> > **Re: Author Response**
> >
> > Thanks for the detailed response!
> >
> > > Although we think the conditions for solving the problem are different, therefore it is difficult to make a fair comparison, they share a common goal to solve OOD tasks as Reviewer BvaU pointed. Therefore, we run experiments for MQL and MIER on MuJoCo tasks, the baselines that provide public reference implementations.
> >
> > I'm glad that the authors performed additional experiments to compare the proposed method to the missing baselines for solving OOD problems in meta-RL, and it seems that the proposed method outperforms these baselines on these environments. Therefore, I will increase my rating to the paper to 5.
> >
> > > As Reviewer BvaU suggested, we present additional experimental results on OOD Humanoid-direction task in Table R6 and Figure R4.
> >
> > It seems that the proposed method does not significantly outperform MQL [1] and MIER [2]. Therefore more environments are needed to show the benefit of the proposed method. Specifically, I suggest the authors to use the mujoco OOD environments proposed in [1, 2] such as Half-Cheetah-Vel-OOD-Medium, Half-Cheetah-Vel-OOD-Hard, Cheetah-Negated-Joints and Ant-Negated-Joints. The code for these environments are public available.
> >
> > Overall, I think the authors' response addressed part of my concerns and therefore I will increase my rating to 5. I encourage the authors to further address my other concerns about OOD environments.
> >
> >
> > References
> >
> > [1] Fakoor, Rasool, et al. "Meta-q-learning." arXiv preprint arXiv:1910.00125 (2019).
> >
> > [2] Mendonca, Russell, et al. "Meta-reinforcement learning robust to distributional shift via model identification and experience relabeling." arXiv preprint arXiv:2006.07178 (2020).

---

> > > ### Author Response · Authors · 2021-08-24
> > > **Additional experiments on suggested OOD environments**
> > >
> > > We are glad that the reviewer found our additional experiments for MQL and MIER on MuJoCo tasks helpful. We appreciate the suggested MuJoCo OOD environments.
> > >
> > > We would like to remind that LDM generates unseen rewards, focusing on MDPs with varying reward dynamics as many previous work (Line 123). Among the four recommended environments, Cheetah-Negated-Joints and Ant-Negated-Joints tasks are first introduced in MIER paper to evaluate extrapolation to OOD state-transition dynamics. We found that LDM is not successful when evaluated on the Negated-Joints tasks. However, our method can work effectively on Half-Cheetah-Vel-OOD-Medium and Half-Cheetah-Vel-OOD-Hard tasks that are introduced to evaluate extrapolation to OOD rewards.
> > >
> > > We import the environments from the MIER repository and evaluate LDM (8 seeds) on Half-Cheetah-Vel-OOD-Medium and Half-Cheetah-Vel-OOD-Hard tasks, where the training target speeds range from 0 to 2.5 and 0 to 1.5, respectively. Both tasks are tested on target speeds ranging from 2.5 to 3.0. We compare LDM to the results of other baselines reported in Figure 4 of the MIER paper. We use the same set of hyperparameters that we used for the Half-cheetah-velocity task (in Section 5.2) except for the extrapolation level $\beta$. Since these tasks deal with extrapolated test tasks, we try multiple values of $\beta \in \\{ 1,2,4,8, 16 \\}$ and find using large $\beta$ to be effective (as discussed in Appendix B), generating tasks with the extrapolated target velocities. We report the results in Table R10 and Figure R7 below.
> > >
> > > **Table R10. Half-cheetah-vel OOD tasks**
> > >
> > > |                    	| Half-Cheetah-Vel-OOD-Medium 	|        	|                    	| Half-Cheetah-Vel-OOD-Hard 	|        	|                    	|
> > > |--------------------	|:---------------------------:	|:------:	|--------------------	|---------------------------	|--------	|--------------------	|
> > > |                    	| Mean                        	| std    	| 95% conf. interval 	| Mean                      	| std    	| 95% conf. interval 	|
> > > | RL2 Oracle (***)        	| -69.172                     	| 15.116 	| 10.475             	| -69.172                   	| 15.116 	| 10.475             	|
> > > | LDM ($\beta$=1.0)  	| -112.098                    	| 23.095 	| 16.004             	| -462.072                  	| 44.581 	| 30.893             	|
> > > | LDM ($\beta$=2.0)  	|  **-74.490**       | 16.179 	| 11.211             	| -453.805                  	| 70.219 	| 48.659             	|
> > > | LDM ($\beta$=4.0)  	| -80.037                     	| 14.195 	| 9.837              	| -372.033                  	| 95.855 	| 66.424             	|
> > > | LDM ($\beta$=8.0)  	| -91.971                     	| 10.374 	| 7.189              	| -251.422                  	| 29.907 	| 20.724             	|
> > > | LDM ($\beta$=16.0) 	| -97.192                     	| 17.081 	| 11.837             	|  **-210.763**                  	| 26.871 	| 18.621             	|
> > > | MIER (*)           	| -82.4                       	| 20.8   	| 23.575             	| -335.1                    	| 73.0   	| 82.577             	|
> > > | MIER-wR (*) (**)   	| -121.8                      	| 44.0   	| 49.770             	| -368.9                    	| 39.2   	| 44.347             	|
> > > | MQL (*)            	| -77.8                       	| 11.6   	| 13.097             	| -387.8                    	| 39.2   	| 44.347             	|
> > > | PEARL (*)          	| -295.4                      	| 13.9   	| 15.717             	| -432.4                    	| 13.5   	| 15.292             	|
> > > | MAML (*)           	| -526.9                      	| 16.2   	| 18.336             	| -532.4                    	| 17.6   	| 19.880             	|
> > >
> > > (*) These numbers are extracted from Figure 4 of the MIER paper.
> > >
> > > (**) A variant of MIER without experience relabeling
> > >
> > > (***) The oracle is trained for target velocities from 0 to 3.0 for both OOD tasks.
> > >
> > > **Link to Figure R7. Half-cheetah-vel OOD tasks)**
> > >
> > > https://drive.google.com/file/d/1A2RTQ-3rzhH6HgAILj14Lx1SZ58qzaW8/view?usp=sharing
> > >
> > > **Half-Cheetah-Vel-OOD-Medium**
> > >
> > > The Half-Cheetah-Vel-OOD-Medium task is relatively easier and requires less extrapolation compared to the Half-Cheetah-Vel-OOD-Hard task. Therefore, it is hard to see a clear difference between LDM, MIER, and MQL as they all perform nearly the same as the oracle. However, it is convincing that LDM with $\beta=2.0$ outperforms all baselines and performs comparably to the oracle.
> > >
> > > **Half-Cheetah-Vel-OOD-Hard**
> > >
> > > We find that LDM with larger $\beta \in \\{8.0, 16.0 \\}$ significantly outperforms MIER and MQL on the  Half-Cheetah-Vel-OOD-Hard task. Since it is a severe extrapolation task, LDM requires $\beta$ large enough to create extrapolated reward dynamics helpful for the test.
> > > The major component of the reward at each step is given as the difference between the agent velocity and the target velocity. On the Half-Cheetah-Vel-OOD-Hard task, even if the agent runs with the velocity of 1.5 during the test (the largest training target velocity), the agent can score approximately -320 = -[2.75 (average test target velocity) - 1.5] * 200 (steps) - 70 (control cost of oracle) return in one episode. Therefore, it is difficult to interpret that the MIER, which is reported to score 335.1 test return, generalized well to the test tasks.
> > >
> > > We hope that the experimental results on Half-Cheetah-Vel-OOD tasks have addressed the reviewer's concerns about OOD environments.

---

### Official Review · Reviewer_Ncyq · 2021-07-16

**Rating:** 6
**Confidence:** 4

**Summary:**

This paper tackles out-of-distribution generalization in meta-reinforcement learning. The proposed method, latent dynamics mixture (LDM), extends meta-RL methods (e.g. variBAD) that support inference and generation of a reward function by interpolating between inferred latents of training tasks to generate new reward functions. The intuition is that "filling in" areas of the task space not supported by the training task distribution corresponds to shifting towards the test task distribution.

**Limitations And Societal Impact:**

This is adequately done.

**Main Review:**

## Originality
The technical contribution is very limited: the delta over variBAD consists entirely of obtaining a convex combination of latent vectors for decoding.

## Quality
The experiments are reasonable, but there is a lack of comparison to methods that also attempt to tackle out-of-distribution generalization, aside from MixReg-based RL2. All other comparisons are made to meta-RL methods that assume iid training and test distributions. Besides quantitative results, informative qualitative analysis that investigates the impact of the proposed method are presented.

## Clarity
The paper is well-written.

## Significance
While this is a cleanly executed paper, I am not sure it will have a significant impact. The technical contribution is undisputably incremental, and the resulting improvements marginal. Given that LDM implicitly assumes that test tasks are "interpolations" of training tasks (this is reflected in the design of the experiments) and that the latent space is regularized by the variational inference training procedure, it's also entirely unsurprising that it works.

## Summary
In essence, my main concern is that the contributions are not sufficient to warrant a NeurIPS publication. However, I look forward to discussing this point with everyone.

## Post-rebuttal update
The authors have clarified via ablations that dropout on the decoder input and separate encoders are important for LDM. They have also expanded their experiments by comparing to meta-learning methods that address out-of-distribution generalization, and on more environments. I have improved my score to a 6.

I still maintain similar concerns to Reviewer 1a3x regarding an implicit "convex hull"/"filling in" assumption underpinning LDM. While the authors have acknowledged this point in their rebuttal, I encourage them to develop this notion further in future revisions. It would make the paper stronger.

**Time Spent Reviewing:**

5

---

> ### Author Response · Authors · 2021-08-10
> **Author Response to Reviewer Ncyq**
>
> We appreciate the constructive advice on emphasizing the contribution and adding baselines for experiments.
>
> **Q1) Limited contribution**
>
> We apologize for not being able to deliver our contribution sufficiently. We have summarized the novelty of our method compared to the existing studies.
>
> **1. vs Recurrence-based standard Meta-RL methods)** We empirically demonstrated that recurrence-based methods such as RL^2 and variBAD fail to solve test tasks strictly disjoint to training tasks. We train the policy with mixture tasks to achieve OOD generalization.
>
> **2. vs off-policy OOD Meta-RL methods)** these methods require trajectories from test tasks before adaptation and a lot of re-labeling or re-training (reusing data collected during meta-training) before adapting to the test task, whereas we provide mixture tasks "during training" and start online adaptation from the very first episode of test task without extra training. Hopefully our response to Q1 (additional empirical comparison to MQL and MIER) of BvaU may address this concern.
>
> **3. vs Task generating (augmenting) methods)** We do not require any parameterization of the environment but meta-learn to encode the environment dynamics into latent values.
>
> We agree that the main delta over variBAD is the mixture training from interpolations of latent vectors. However, the interpolation alone cannot improve the test returns for unseen tasks. We require dropout on the state input of the decoder and separate encoders for the decoder and the policy network. We emphasize the necessities for both mixture AND dropout in detail with ablation studies in responses to Q5 of Reviewer 1a3x.
>
> **Q2) Lack of comparison to tackle out-of-distribution generalization.**
>
> To address the concern, we provide additional experimental results. We present comparison to recent off-policy OOD meta-RL methods: Meta Q-learning (MQL) and Model Identification and Experience Relabeling (MIER). In our response to Q1 for Reviewer BvaU, we explain the different problem settings of LDM and these methods, then report the empirical results on MuJoCo tasks, and show that LDM consistently outperforms these methods.
>
> **Q3) Design of experiments as "interpolations" of training tasks**
>
> The goal of our work was to improve generalization performance on out-of-distribution tasks of standard benchmarks. We agree that more general task formulation beyond the standard interpolation setting is certainly an interesting topic. However, at current state, we have confirmed experimentally that many meta-RL methods, including OOD-targeted methods, have trouble solving unseen interpolated tasks in the standard benchmarks. We empirically demonstrate that LDM can outperform the baselines and generalize better in such situations. In an effort to extend our method from a simple interpolation, we show in Appendix B.2 that LDM can be helpful for extrapolation task situations when we use larger $\beta$ such as 2 or 2.5.

---

### Official Review · Reviewer_yAYE · 2021-07-17

**Rating:** 7
**Confidence:** 4

**Summary:**

The authors attempt to solve a problem in context-based RL, where a model tries to learn and identify the context within which it is behaving. This context can be defined by differences between MDPs, such as differences in the transition dynamics, and the reward function; the agent will make use of such a context in its policy.

This work builds heavily on the variBAD, as is mentioned in the paper. As its main contribution, it introduces a so-called Latent Dynamics Mixture model where during training, latent context values that are seen over the sampled training tasks are used (additively) to generate new, 'imaginary' tasks, to augment the training.

This method aims to improve on the problem of generalising to unseen tasks that are generated from an unseen task distribution.


**Limitations And Societal Impact:**

I think that some limitations are mentioned (for example the fact that dropout is important for the imaginary tasks to actually look at unseen goals). I think some more discussion of why this is would have been good. See Main Review for more details about this.


**Main Review:**

### Originality
- This paper builds on variBAD, but I believe has sufficient novelty. I haven't seen similar work otherwise.


### Quality
- I think that experiments that the authors had run, do support their claim that generalisation to unseen tasks from an unseen distribution works better with their models. But some questions do arise. See (Misc. Comments for some questions I have).

- Are 4 seeds really enough for any meaningful statistical significance?

### Clarity
- I think that this paper is written sufficiently well. I would have liked to have seen the equation of the loss they are using, written in terms of their notation. It would help with my understanding of what is going on, and the interaction between the different components of this.

- That being said, Figure 2 was quite a nice summary of their solution.

- I appreciate the inclusion of the code, and what appears like sufficient hyperparameter details of how to reproduce this work in the Appendices.

- Line 40: I think that the language here could be clarified a little. When I first read this, I thought that the policy would be able to learn new 'actions' for its policy. Perhaps more clarity about which environment this is in, and more precise descriptions would be useful.

- Line 249: Repeated definition of H+ is repeated as in Line 161. Also, I'm not certain H+ is described or used anywhere substantial.

- I enjoyed Figure 6(b) and (c), as they are nice visualisations the different ways the methods behaved.

### Significance
- I think that this work is quite interesting. This could be a nice solution for OOD problems in RL as discussed in the conclusion of the paper.

  But, some questions arise from this work (see Misc. Comments below) that I think should be addressed first.

### Misc. comments
- Generally speaking, I am confused about the use of the word "Dynamics" in LDM. Is this referring to the dynamics of the MDP? Or is it referring to some dynamics of the latent context variable?

-  I am also confused as to how exactly this method would help generalise to new tasks. As I understand it, a main novelty is the generation of the imaginary tasks, which are generated as a convex combination of the latent context values from the training tasks. Now, if the claim is that methods that don't use such a generation of imaginary tasks do not generalise well because the latent values are generated only over $M_{train}$, why is there an expectation that a convex sum of such $m$ would result in meaningfully new tasks?

- In fact, from Figure 4, and the description given in the paragraph starting at Line 268, it seems like the dropout plays a key role in generalising to unseen goals. So do the imaginary tasks do what is claimed?

   From the previous submission to ICML 2021, the authors had added Figure 4 to address the following issue:

   "We included additional empirical results regarding the tasks generated by LDM in Figure 4. We added a paragraph describing that LDM indeed generates meaningful new tasks that help to solve unseen test tasks."

   I'm not certain the new figure answers this, due to the effects of dropout.

- What is the significance of encoders for the policy network and latent dynamics network not being shared? If I understand correctly, the policy uses a latent embedding too yes? Couldn't it use the embedding learnt from Encoder-v? Why are they separated? Could there be a comparison where the encoders are shared, as an ablation?

### Conclusion
I think that this work is interesting. The clarifications of the questions above would help my confidence in the work.

**Time Spent Reviewing:**

6

---

> ### Author Response · Authors · 2021-08-10
> **Author Response to Reviewer yAYE**
>
> Thank you for the thorough review and for acknowledging the sufficient novelty of our work.
>
> **Q1) Are 4 seeds really enough for any meaningful statistical significance?**
>
> We increased the number of seeds from 4 to 8. Please refer to our response for the same concern at Q1 of Reviewer 1a3x.
>
> **Q2) Equation of the loss in terms of our notation**
>
> We paraphrase the objective functions in variBAD in terms of our notation.
>
> **1. Normal workers** train the policy network and latent dynamics network separately in $M_\textrm{train}$, by maximizing following objectives.
>
> Policy Network: $ \mathcal{L}1(\phi_p, \psi)=E_{p(M_{\textrm{train}})} [\mathcal{J(\phi_p, \psi)}]$
>
> Latent Dyanmics Network  (VAE): $ \mathcal{L}2(\phi_v, \theta_R)=E_{p(M_{\textrm{train}})} [ELBO_{t}(\phi_v, \theta_R)]$
>
> (1) The ELBO term
> $$ELBO_{t}(\phi_v, \theta_R)= E_{\rho(M,\tau_{:H^{+}})}[E_{q_{\phi_{v}}(m|\tau_{:t})}[\log p_{\theta_R}(\tau_{:H^{+}}|m,a_{:H^{+}-1})]-KL(q_{\phi_{v}}(m|\tau_{:t})||p_{\theta_{R}}(m))] \leq E_{\rho(M,\tau_{:H^{+}})}[\log p_{\theta_R}(\tau_{:H^{+}}|a_{:H^{+}-1})]$$
> $\rho(M,\tau_{:H^{+}})$ is the trajectory distribution.
>
> (2) The policy term
> $ \mathcal{J(\phi_p, \psi)} = E_{b_{0},\phi_{p},\psi}[\sum_{t=0}^{H^+-1} \gamma^{t}R^{+} (r_{t+1}|s^+_t,a_t, $
>
> $s^+_{t+1})]$, where
>
>
> $s^+_t=(s_t,b_t)$,
>
> $R^+=E_{b_{t+1}}[R(s_t,a_t,s_{t+1})]$ ...(*)
>
>  **2. A mixture worker** only trains the policy network in $\hat{M}
>
> Policy Network: $ \mathcal{L}3(\phi_p, \psi)=E_{p(\hat{M})} [\mathcal{J(\phi_p, \psi)}]$
>
> The policy objective is same as (2) with reward term $R(s_t,a_t,s_{t+1})$ in (*) replaced by the output of the decoder conditioned on the mixture latent model $\hat{m}$, i.e., $R(s_t,a_t,s_{t+1})\leftarrow p_\{\theta_{R}}(s_t,a_t,s_{t+1};\hat{m})$
>
> **Q3) Clarity about the environment in Line 40**
>
> We apologize for the confusion. We will clarify that LDM creates new goal directions for the policy to learn.
>
> **Q4) Repeated definition of $H^+$ in Line 161**
>
> $H^+$ is used in Figure 2 and in the caption of Figure 4. It is also used in the response to Q2. We will remove the redundant definition of it.
>
> **Q5) Confusion about the use of the word "Dynamics in LDM.**
> We apologize for the confusion. What we mean by "latent dynamics model" is the value of $m$ itself, which is the latent embedding of the environment's dynamics, not the dynamics of the latent variable. We will include a precise definition of the term to clarify.
>
> **Q6) Now if the claim is that methods that don't use such a generation of imaginary tasks do not generalise well because the latent values are generated only over $M_{\textrm{train}}$, why is there an expectation that a convex sum of such m would result in meaningfully new tasks?**
>
> Our response to Q3 and Q5 of Reviewer 1a3x may help address this concern. Conventional methods without imaginary tasks can also encode contexts for unseen but similar tasks. As mentioned in Q2 and Line 145, LDM also does not train the latent dynamics network (the VAE part) for mixture tasks, but it trains only the policy part for mixture tasks. Still, the latent values for the unseen tasks can be well encoded (as the convex sum of seen training latent values) as can be seen in Figure 12(b) and 12(d) in Appendix E. However this encoding is possible only under the premise that the policy is prepared for exploration to find out sufficient information to infer the task (e.g., explore the regions in $M_{\textrm{test}}$ of gridworld efficiently until it discovers a rewarding state). By training the policy in such mixture environment,  the policy can learn the exploration strategy to reach such latent values and learn to exploit the task once it is inferred.
>
> **Q7) In fact, from Figure 4, and the description given in the paragraph starting at Line 268, it seems like the dropout plays a key role in generalising to unseen goals. So do the imaginary tasks do what is claimed?**
>
> Our response to Q5 of Reviewer 1a3x may help address this concern. From ablation results and the motivating example, it can be seen that using dropout only has no significant effect. By using both dropout and mixture training, we could create reward maps such as the ones in Figure 4c), 4d) and 4e), where the maximum reward is attained at a state in $M_{\textrm{test}}$. Such mixture task induces explorations to the regions in $M_{\textrm{test}}$. As explained in our response to Q5 of Reviewer 1a3x, if we only use dropout without mixture, some nearby states of $M_{\textrm{train}}$ can return positive rewards. However, the maximum reward is likely to be attained in one of the goal states of $M_{\textrm{train}}$.
>
> **Q8) Separated encoders.  If I understand correctly, the policy uses a latent embedding too yes? Couldn't it use the embedding learnt from Encoder-v? Why are they separated? Could there be a comparison where the encoders are shared, as an ablation?**
>
> Yes, the policy network also uses latent embedding as in Figure 2. Encoder-p is trained for both true and mixture tasks, whereas the encoder-v (encoder for VAE part) is trained for true tasks only. We encountered instability in policy training when the encoding, trained for decoder with dropout, was used for policy. If we have separate structure, even when the VAE is not perfectly trained due to the dropout, it does not affect the policy. We report the ablation results requested in the following anonymous link. This question is related to Q2 of reviewer 1a3x about variBAD with dropout.
>
> **Link to Figure R2. LDM Ablations on Gridworld)**
>
> https://drive.google.com/file/d/1zVA72OO2S8O-hS_j0a2VbO_QMA8W-9XJ/view?usp=sharing
>
> Also, there is an advantage to having the structures of the policy network and latent dynamics network completely separated. Instead of using the RL^2 policy we used, we easily replace it with other RL architectures such as the ones mentioned in Q1 of Reviewer BvaU.

---

### Official Review · Reviewer_1a3x · 2021-07-19

**Rating:** 5
**Confidence:** 3

**Summary:**

This work studies the partial observability problem of training a policy to adapt to new, unseen out-of-distribution tasks at test time. The authors focus on the setting where the dynamics across all tasks are the same but reward changes. They propose to train an ensemble of latent dynamics models, policies, and reward decoders and train an additional mixture version with dropout. During training, they replace the real reward with an imaginary reward generated by the mixture decoder to create new imaginary tasks. They show that in gridworld and MuJoCo environments, their method generalizes better than other Meta-RL methods.


**Limitations And Societal Impact:**

Yes.

**Main Review:**

To really evaluate these results requires more than 4 seeds. The variance seems quite high in Figure 3. Given the importance of dropout to prevent memorization of train tasks, another good baseline to check would be RL^2 and/or VariBAD with dropout as well. Perhaps that is all the regularization needed to get better generalization performance? You mention that LDM test time behavior is to first try all the train goal points before exploring for new test goal points. Given that the other recurrence-based baselines also have > 0 reward at test time, do they do something similar?

Overall it seems like a nice idea but I’m not convinced by the results. More analysis of the types of problems that LDM would work well vs. not well for would also be appreciated. All the examples chosen are of tasks with more “continuous” context spaces. I assume LDM fails in tasks with only discrete structure? What types of test tasks would this fail at? It seems like there is some notion of a convex hull over training environments that this captures. Requiring a strict separation of train and test seems like a limitation? Does it need to be strict? Does the method not work if it is not?

I am currently giving this work a borderline reject which I am happy to raise to an accept contingent on some analysis and discussion of limitations and assumptions being made about the structure of the environment and results with 10 seeds.

**Time Spent Reviewing:**

3

---

> ### Author Response · Authors · 2021-08-10
> **Author Response to Reviewer 1a3x (2/2)**
>
> **Q4) More analysis of the types of problems that LDM would work well vs. not well.**
>
> LDM can be effective when the test tasks are similar to the mixture tasks generated by LDM, where the pool of mixture tasks is controlled by $\beta$ and $p_{drop}$. Additional analysis in Appendix B.1 and B.2 may address this concern. Referring to Figure 9 in Appendix B, LDM with $\beta=1.5$ can be effective if the test tasks are in $M_{\textrm{test1}}$, however it will not work well for tasks in $M_{\textrm{test2}}$. But this is a problem that occurs in other methods such as RL^2 and variBAD as well. LDM with large extrapolation parameter $\beta=2.5$ can effectively solve tasks in $M_{\textrm{test2}}$.  An interesting future work could be creating curriculums for various $\beta$ to prepare for more general pools of mixture tasks that are likely to appear during testing.
>
>
> **Q5) All the examples chosen are of tasks with more “continuous” context spaces. I assume LDM fails in tasks with only discrete structure?**
>
> Yes, LDM would fail in tasks with discrete structures if it does not use dropout for the input state of the decoder. The gridworld example has discrete task space, therefore without dropout, the tasks generated as interpolations will not be able to generalize reward for states in $M_{\textrm{test}}$ (as in Figure 4a). This failure in discrete structure can be confirmed in experiment results of LDM without dropout ($p_{drop}=0$) in Figure 11 of Appendix D. Dropout alleviates this problem by smoothing the state-to-reward function.
>
> An extreme intuitive example would be a one-dimensional goal reaching task, where there are two tasks with discrete reward functions: $R_1(x)=\delta(x)$ and $R_2(x)=\delta(x-1)$. $x$ is the raw coordinate observation and assume the agent can move along the $x$-axis. Then the decoder will be trained to estimate the reward dynamics of the first and second task as $p_{\theta_R}(s(x);m^{(1)})=\delta(x)$ and $p_{\theta_R}(s(x);m^{(2)})=\delta(x-1)$, respectively. Where $s$ is a state embedding function.
>
> -	(**Mixture without Dropout**) Even if we mix the tasks as $\hat{R}(x)=\alpha_1\delta(x)+\alpha_2\delta(x-1)$, we can not augment a non zero reward besides $x=0$ or $x=1$.
> -	(**Dropout without Mixture**) For simplicity, we may assume augmented reward functions as gaussian function $g(x; \mu, \sigma^2)$ when adding dropout, i.e., $p_{\theta_R}(\tilde{s}(x);m^{(1)})= g(x; 0,\sigma^2)$ and $p_{\theta_R}(\tilde{s}(x);m^{(2)})= g(x; 1,\sigma^2)$, where $\tilde{s}(x)$ is the state embedding of raw observation x with dropout. The dropout makes the reward functions smoothed to allow state other than 0 and 1 to attain non-zero reward. The spread ($\sigma$) of the reward function will become larger as the dropout rate $p_{drop}$ increases. However, without mixing, it is highly likely that the maximum reward will be attained near $x=0$ or $x=1$.
> -	(**Mixture and Dropout**)
> Let us generate an augmented task with reward function $\hat{R}(x)=\alpha_1 g(x; 0,\sigma^2) + \alpha_2 g(x; 1,\sigma^2)$. With sufficiently large $\sigma$ (large $p_{drop}$), we can create task that attains the largest reward for $x$ between 0 and 1.
>
> **Link to Figure R5. Visualization of the intuitive example)**
> https://drive.google.com/file/d/1csA00eQPA8iXsKrqievEc73XIiNRzCLc/view?usp=sharing
>
> Therefore, both dropout AND mixture training are essential elements to create general tasks. The reward map in Figure 4 can be regarded as an expansion of this example in 2-dimension, where a task with maximum reward in $M_{\textrm{test}}$ can be created (related to Q7 of Reviewer yAYE). This analysis on discrete context space provides insights on why using a larger dropout rate for Gridworld ($p_{drop}=0.7$) than MuJoCo ($p_{drop}=0.5$) was effective.
>
> **Q6) Convex hull over training environments.**
> This response is related to Q4. It is true that there is some notion of a convex hull over training environments as in Figure 12 of Appendix E. In fact, if the latents of the test tasks exist in this convex hull, the decoder can create meaningful mixture tasks with $\beta=1.0$. Even if it is not inside the convex hull, it is possible to extrapolate even outside the hull to some extent if $\beta$ is well adjusted as in Figure 9 of Appendix B.2.
>
> **Q7) Requiring a strict separation of train and test seems like a limitation? Does it need to be strict? Does the method not work if it is not?**
>
> The experimental setup with separated train and test was established to expose the limitations of existing methods with i.i.d. task sampling. LDM would still work even the separation is not strict, but the gain of LDM compared to existing methods will be minimized if the test tasks become similar to training tasks.

---

> ### Author Response · Authors · 2021-08-10
> **Author Response to Reviewer 1a3x (1/2)**
>
> We appreciate the reviewer for suggesting efficient ways to clearly analyze the advantages and limitations of the proposed method.
>
> **Q1) Using more number of seeds**
>
> The use of 4 random seeds followed the related studies that use 3~5 random seeds for continuous control tasks: RL^2(3), ProMP(3), Pearl(3), variBAD (20 for gridworld and 5 for mujoco), MQL(5), MIER(3), FLAP(3). Nevertheless, we strongly agree with Reviewer 1a3x that a new standard needs to be established in the field of meta-RL for reliable experimental results. Therefore, we double the number of seeds used from 4 to 8 to report our experimental results.  We report the final test returns in Gridworld below in Table R1.
>
> **Table R1. Gridworld 8 Seeds, test return at the N-th episode**
>
> |                  | 4 seeds |       |                   |   | 8 seeds | 　    |                   |
> |------------------|:-------:|:-----:|-------------------|---|--------|-------|-------------------|
> |                  | mean    | std   | 95% conf. interval |   | mean   | std   | 95% conf. interval |
> | LDM              | 14.512  | 3.628 | 3.555             |   | 14.062 | 3.150  | 1.782             |
> | Mixreg           | 7.320   | 3.482 | 3.412             |   | 8.644  | 4.451 | 2.518             |
> | RL^2              | 6.882   | 3.138 | 3.075             |   | 6.333  | 3.341 | 1.890             |
> | RL^2 Oracle       | 24.294  | 0.353 | 0.346             |   | 24.024 | 0.833 | 0.471             |
> | variBAD          | 3.021   | 3.012 | 2.952             |   | 2.045  | 3.734 | 2.112             |
> | variBAD   Oracle | 23.316  | 1.315 | 1.289             |   | 22.775 | 1.478 | 0.836             |
> | ProMP            | -2.636  | 0.386 | 0.378             |   | -2.582 | 0.363 | 0.205             |
> | E-MAML           | -2.627  | 0.298 | 0.292             |   | -2.818 | 0.119 | 0.067             |
>
> We share an anonymous link to the new graph that corresponds to Figure 3 of the main manuscript with 8 seeds.
>
> **Link to Figure R1. Gridworld 8 seeds)**
>
> https://drive.google.com/file/d/1VMMIY68UjdLZsCCBAchfReGtTdYCvDOQ/view?usp=sharing
>
> Referring to Figure R1 and Table R1, there are slight changes in the mean and std between the results with 4 random seeds and 8 random seeds. The fluctuation over timestep has been greatly reduced in figure R1, achieving more stable convergence. 95% confidence interval is also reduced for most results.
>
> Due to the time constraints, additional experiments on the MuJoCo environments with more seeds are currently in progress. We are going to update our comment as soon as we obtain the remaining results for baselines with 8 seeds. At the moment, we report the LDM performance with 8 random seeds.
>
> **Table R2. MuJoCo 8 seeds, test return at the N-th episode**
>
>
> |                 | 4 seeds   |        |                   | 8 seeds  | 　      |                   |   |
> |-----------------|----------|--------|-------------------|---------|---------|-------------------|---|
> |                 | mean     | std    | 95% conf. interval | mean    | std     | 95% conf. interval |   |
> | LDM Ant-dir     | 1042.517 | 59.736 | 58.543         | 1002.510 | 78.633 | 54.490|   |
> | LDM Ant-goal    | -221.717 | 48.319 | 47.353          | -213.593 | 36.213 | 25.094           |   |
> | LDM Cheetah-vel | -141.597 | 26.259 | 25.734          | -143.682 | 23.378 | 16.200           |   |
>
> There was an outlier out of the four additional runs of Ant-dir task with a final return of 738. This resulted in a decrease in the mean, but still, LDM outperforms all other baselines in all tasks.
>
> **(updated on August 17th)** We have completed additional 4 runs for the remaining baselines on MuJoCo tasks. We report the final result in our common comment for all reviewers.
>
> **Q2) RL^2 and/or VariBAD with dropout as well**
>
> This is a great point. RL^2 does not have a decoder, therefore we do not have to consider the situation where the context $m$ is ignored. However, RL^2 with dropout on the state input of the policy is evaluated in Appendix D Figure 11 (RL^2 dropout). RL^2 dropout cannot be trained stably, since training policy network with a multi-step policy gradient loss is much more complex than training the decoder with a single step regression loss (described at Line 154). We run additional experiments for variBAD with dropout on the state input of decoder (same as LDM) and report the result in Table R3 and Figure R2.
>
> **Table R3. Gridworld Ablations, test return at the N-th episode**
>
> |                    | mean   | std   | 95% conf. interval |   |   |   |   |
> |--------------------|--------|-------|--------------------|---|---|---|---|
> | LDM                | 14.062 | 3.150 | 1.782              |   |   |   |   |
> | LDM no dropout     | 8.830   | 3.444 | 2.755              |   |   |   |   |
> | LDM shared encoder | -2.285 | 0.711 | 0.569              |   |   |   |   |
> | RL^2                | 6.333  | 3.341 | 1.890              |   |   |   |   |
> | RL^2 dropout        | -0.329 | 0.599 | 0.479              |   |   |   |   |
> | variBAD            | 2.045  | 3.734 | 2.112              |   |   |   |   |
> | variBAD dropout    | -2.342 | 0.948 | 0.758              |   |   |   |   |
>
> **Link to Figure R2. LDM Ablations on Gridworld)**
>
> https://drive.google.com/file/d/1zVA72OO2S8O-hS_j0a2VbO_QMA8W-9XJ/view?usp=sharing
>
> VariBAD shares an encoder for both VAE and the policy network and does not backpropagate the policy loss to the encoder. Therefore, the encoding, trained with decoder with dropout, may disturb the policy network from stable training. For this reason, we separate the encoders for LDM to prevent the latent dynamics network from interfering with the policy network.
>
> **Q3) Recurrence-based baselines with >0 reward at test time, do they do something similar?**
>
> This is a key question in explaining the difference between LDM and other recurrence-based baselines (RL^2 and variBAD). The answer is ‘yes but not always and not optimally.
> Both LDM and these baselines search for goals in $M_{\textrm{train}}$ first. The key difference occurs during the test after the agent discovers that there are no goals in $M_{\textrm{train}}$. LDM starts to explore for the regions in $M_{\textrm{test}}$ efficiently, whereas other methods revisit the states in $M_{\textrm{train}}$  or randomly explore nearby states in $M_{\textrm{test}}$. Therefore other recurrence methods have less chance to discover test goals(>0 rewards) within the limited horizon.
> Once they discover the goal by chance, they can exploit (but not always and not with the shortest path) the test goals. This can be seen from variBAD and RL^2 achieving returns greater than 0 for some tasks as in Figure 1c. To quantitatively evaluate this behavior, we report the average number of test goals(r>0) discovered at least once until the (N-1)th episode and the number of times the agent succeeds to revisit previously discovered goals at the N-th episode.
>
> **Table R4. Gridworld Revisit Count for r>0, mean of 8 runs**
>
>
> |         | Number of goals discovered (out of 27 test tasks)  at least once until the (N-1)th episode | Number of goals that are discovered until  the (N-1)th episode and revisited at the N-th episode |
> |---------|----------------------------------------------------------------------------------------------|---------------------------------------------------------------------------------------------------|
> | LDM     | 17.75                                                                                        | 16.75                                                                                             |
> | RL2     | 10.625                                                                                       | 10                                                                                                |
> | variBAD | 11                                                                                           | 5.375                                                                                             |
>
> First, LDM discovers more test goals than RL^2 and variBAD until the (N-1)th episode due to efficient exploration. LDM and RL^2 revisit most of the previously discovered goals in the last episode. VariBAD succeeds to revisit half of the once-discovered goals, resulting in low test returns at the N-th episode.

---

> ### Author Response · Authors · 2021-09-02
> **A kind reminder for Reviewer 1a3x**
>
> Dear reviewer 1a3x,
>
> Thank you for your effort to review our work.
>
> We would like to kindly remind you to check our author response if you have not already. We would appreciate it if you could inform us whether our response successfully addressed your concerns. Even a short statement (why our response did or didn't change your assessment of our work) would be of great help for us. Thank you!

---

### Author Response · Authors · 2021-08-10
**Summary of the Author Response**

We thank the reviewers for their time and effort to provide constructive comments. We summarized the author response we prepared over the past week to address the concerns raised by the reviewers.

- Experimental results with more number of seeds (Reviewer 1a3x and yAYE).
- Ablation results and a motivating example that illustrate the importance of using both mixture **AND** dropout (Reviewer 1a3x, yAYE and Ncyq).
- Experimental results including other off-policy OOD meta-RL methods and comparison to LDM (Reviewer NCyq and BvaU).
- Additional experimental results on OOD Humanoid-dir task (Reviewer BvaU).

The detailed responses and answers to other questions are written in separate comments. We believe we can greatly strengthen the significance of our study by reflecting on the reviewers' comments. If there are any questions and concerns that have not been fully addressed, please post a comment at any moment for us to respond.

*The numbers in all tables are results measured at the end of training (numbers representing the last point of each plot).

---

> ### Author Response · Authors · 2021-08-17
> **MuJoCo Experiment Results with 8 Random Seeds**
>
> We have completed additional runs (4 more random seeds) of the three OOD MuJoCo tasks (Ant-dir, Ant-goal, Cheetah-vel). We report the final test returns at the N-th episode in Table R7, R8, R9, and Figure R6 (corresponding to Figure 5 in the main manuscript).
>
> As the number of seeds is increased, there are slight changes in the final mean and std, but it can be confirmed that LDM still outperforms the existing non-oracle methods.
>
> **Table R7. Ant-dir 8 seeds**
>
> | Ant-dir        |  4 seeds  |         |                    |   8 seeds  |         |                    |
> |----------------|:--------:|:-------:|--------------------|:--------:|:-------:|--------------------|
> |                | mean     | std     | 95% conf. interval | mean     | std     | 95% conf. interval |
> | LDM            | 1042.517 | 59.736  | 58.541             | 1002.510 | 78.633  | 54.490             |
> | Mixreg         | 629.710   | 80.274  | 78.669             | 674.007  | 98.662  | 68.369             |
> | RL2            | 707.622  | 101.458 | 99.429             | 718.606  | 111.101 | 76.989             |
> | RL2 oracle     | 1005.489 | 180.743 | 177.128            | 1061.118 | 140.794 | 97.565             |
> | variBAD        | 841.532  | 51.923  | 50.885             | 721.465  | 205.520 | 142.418            |
> | variBAD oracle | 1101.936 | 83.721  | 82.047             | 1061.118 | 140.794 | 68.369             |
> | proMP          | 349.316  | 7.812   | 7.656              | 358.780  | 29.878  | 20.704             |
> | E-MAML         | 393.623  | 20.458  | 20.049             | 378.291  | 30.003  | 20.791             |
> | PEARL          | 493.195  | 83.937  | 82.258             | 507.379  | 105.677 | 73.231             |
> | MQL            |          |         |                    | 444.328  | 26.232  | 18.178             |
> | MIER           |          |         |                    | 893.876  | 45.589  | 31.592             |
>
>
> **Table R8. Ant-goal 8 seeds**
>
> | Ant-goal       |  4seeds  |        |                    | 8 seeds  |        |                    |
> |----------------|:--------:|:------:|--------------------|----------|--------|--------------------|
> |                | mean     | std    | 95% conf. interval | mean     | std    | 95% conf. interval |
> | LDM            | -221.717 | 48.319 | 47.353             | -213.593 | 36.213 | 25.094             |
> | Mixreg         | -346.954 | 97.081 | 95.139             | -320.207 | 88.857 | 61.575             |
> | RL2            | -260.179 | 42.111 | 41.269             | -269.488 | 50.995 | 35.338             |
> | RL2 oracle     | -221.881 | 42.795 | 41.939             | -210.330 | 37.238 | 25.805             |
> | variBAD        | -226.793 | 54.182 | 53.098             | -247.913 | 69.412 | 48.100             |
> | variBAD oracle | -156.586 | 60.629 | 59.416             | -156.691 | 60.170 | 41.696             |
> | proMP          | -335.101 | 5.253  | 5.148              | -329.848 | 6.937  | 4.807              |
> | E-MAML         | -330.762 | 10.812 | 10.596             | -331.768 | 10.550 | 7.311              |
> | PEARL          | -222.25  | 23.381 | 22.913             | -235.495 | 34.507 | 23.912             |
> | MQL            |          |        |                    | -353.330 | 19.546 | 13.545             |
> | MIER           |          |        |                    | -255.415 | 16.318 | 11.308             |
> **Table R9. Cheetah-vel 8 seeds**
>
> | Cheetah-vel    |  4 seeds  |        |                    | 8 seeds   |        |                    |
> |----------------|:--------:|:------:|--------------------|----------|--------|--------------------|
> |                | mean     | std    | 95% conf. interval | mean     | std    | 95% conf. interval |
> | LDM            | -141.597 | 26.259 | 25.734             | -143.682 | 23.378 | 16.200             |
> | Mixreg         | -259.531 | 27.575 | 27.024             | -251.257 | 24.083 | 16.689             |
> | RL2            | -205.074 | 93.579 | 91.707             | -214.412 | 73.612 | 51.011             |
> | RL2 oracle     | -63.907  | 28.655 | 28.082             | -58.607  | 23.510 | 16.292             |
> | variBAD        | -200.626 | 26.991 | 26.451             | -195.312 | 28.005 | 19.406             |
> | variBAD oracle | -34.345  | 7.595  | 7.443              | -44.920  | 24.269 | 16.818             |
> | proMP          | -192.701 | 6.124  | 6.002              | -186.393 | 10.895 | 7.550              |
> | E-MAML         | -224.906 | 8.499  | 8.329              | -229.564 | 8.815  | 6.108              |
> | PEARL          | -265.142 | 7.784  | 7.628              | -286.836 | 17.677 | 12.250             |
> | MQL            |          |        |                    | -198.099 | 20.242 | 14.027             |
> | MIER           |          |        |                    | -192.678 | 24.452 | 16.944             |
>
> **Link to Figure R6)**
>
> https://drive.google.com/file/d/1bEQReyp8xv0PXM7qbE1Fnrv_moGxr_6b/view?usp=sharing
>
>
> The standard deviation may increase or decrease when the number of seeds is increased. This is not only because of the high variance problem of RL but also because of outlier seeds, which often occur due to the nature of the OOD task. However, we obtained more significant experimental results as the 95% confidence interval decreased for most results.

---

### Decision · Program_Chairs · 2021-09-27

**Decision:**

Accept (Poster)

**Comment:**

This paper proposes to learn an ensemble of latent dynamics models and provide a mixture of them as input to a reward decoder, which allows the entire framework to generate imaginary tasks (i.e., imaginary reward functions). By training the agent on the generated tasks, the agent can generalize better to out-of-distribution tasks on grid-world and MuJoCo environments. The idea of generating imaginary reward functions using a mixture of latent dynamics model is novel, though the overall framework builds upon VariBad. Although there was an initial concern around the significance of the results (e.g., lack of baselines, challenging environments), the additional results during the rebuttal period addressed most of the questions raised by the reviewers. However, the question about how the proposed framework can extrapolate so well and how the mixture hyperparameters ($\beta$) affects interpolation/extrapolation performance remains the same, which may need to be discussed in the main paper rather than in the appendix. Given that the main result is still solid and consistent, I'd recommend acceptance for this paper.